# Mechanistic insights into recruitment and regulation of the RNA helicase UPF1 in replication-dependent histone mRNA decay

Alexandrina Machado de Amorim[1,14], Guangpu Xue[1,10,14], Wenxia He [2,3], Theresa Dittmers [1,11], Sarah Lewandowski [1], Cecilia Perez-Borrajero [4,12], Juliane Bethmann[5,6], Nevena Mateva[1], Clemens Krage[1], Vidhyadhar Nandana [1,13], Bernhard Loll [7], Tarek Hilal [7,8], Janosch Hennig [4,9], Henning Urlaub [5,6], William F. Marzluff [2,3] & Sutapa Chakrabarti [1] ✉

Metazoan histone mRNAs are a unique class of mRNAs that lack the poly(A) tail present in all other eukaryotic transcripts. Instead, they end in a conserved stem-loop (SL) structure, necessitating a decay mechanism that is distinct from deadenylation-initiated degradation. Here, combining structural and functional approaches, we elucidate molecular mechanisms of initiation of histone mRNA decay. At the end of S-phase, the RNA helicase UPF1, the exoribonuclease 3'hExo and stem-loop binding protein SLBP all contribute to histone mRNA degradation, although how they are mechanistically coupled remained unknown. The cryoEM structure of an UPF1:SL RNA complex, presented here, shows that binding of UPF1 partially melts the RNA stem in the absence of ATP, harnessing the free energy derived from RNA-binding to unwind RNA. This melting event primes the SL-RNA for decay by 3'hExo. Using biochemical and cellular analyses, we demonstrate that SLBP directly engages the UPF1 helicase core to attenuate its unwinding activity and prevent premature degradation. Activation of UPF1 at a later stage promotes SL-RNA decay. We provide direct evidence that UPF1, SLBP and 3'hExo form a degradosome-like assembly that functionally couples SL unwinding and degradation, highlighting a dynamic and intricate network of UPF1-centric interactions that orchestrates timely histone mRNA decay.

Expression of histone proteins in metazoans is coupled with DNA replication to ensure that newly synthesized DNA is efficiently packaged into chromatin. This coordinated expression is achieved by rapid upregulation of replication-dependent (RD) histone mRNAs in the DNA synthesis phase (S-phase) of the cell cycle, and an equally rapid decrease when cells exit the S-phase[1,2]. Metazoan RD-histone mRNAs are unusual in that they contain a stem-loop (SL) structure at their 3' end, instead of the poly(A) tail that is present in all other eukaryotic

cellular mRNAs[3,4]. The SL binding protein (SLBP) binds at the 5' side of the SL during transcription and remains associated with the histone SL RNA until its decay[5,6]. Decay of the histone mRNA is initiated at its 3' end by the concerted action of SLBP, the 3'−5' exoribonuclease 3'hExo (also known as ERI1) and the RNA helicase UPF1, which is involved in multiple pathways of cytoplasmic mRNA decay[7–11]. Structural studies of the histone SL RNA bound to SLBP and 3'hExo showed that the two proteins stably bind the SL on opposite sides but do not directly

interact with each other[12,13]. Despite the active nuclease, the SL remains intact in the ternary SLBP-SL-3′hExo complex and is only slightly shortened in the binary SL-3′hExo complex, where the RNA cannot reach the nuclease active site. These observations suggest that the SLBP-bound SL is resistant to spontaneous degradation and must undergo a change in conformation that makes it amenable to degradation by 3′hExo. Two lines of evidence suggest a pivotal role for UPF1 in RD histone mRNA decay: an iCLIP analysis showed that UPF1 is recruited to the 3′-untranslated region (UTR) of RD histone mRNA, immediately 5′ of the SLBP-binding site in the absence of ribosomes, and knockdown of UPF1 or overexpression of a mutant incapable of binding ATP (UPF1 K498A) slows histone mRNA degradation in cells[8,14]. Additionally, deletion of domains of UPF1 that regulate its catalytic activity through intra–molecular interactions slow down histone mRNA decay[15,16]. Indeed, UPF1 was shown to immunoprecipitate with SLBP in an RNA-independent manner, after inhibition of DNA replication, accounting for its precise binding site on RD-histone mRNA.

UPF1 is an essential protein in mammalian cells and is a member of the superfamily 1B (SF1B) of RNA helicases that translocate on single-stranded nucleic acid stretches in the 5′–3′ direction[17–19]. Translocation of the helicase is dependent on ATP hydrolysis and mediates unwinding of structured RNA and remodeling of ribonucleoprotein complexes (RNPs). The role of UPF1 in mRNA decay has been most investigated in the context of nonsense-mediated mRNA decay (NMD), where it was shown to use its RNP-remodeling activity to disassemble mRNPs undergoing NMD[20,21]. More recently, the ATPase activity of UPF1 was found to be essential for accurate NMD target selection and therefore, critical for progression of NMD[22,23]. The catalytic activity of UPF1 is stringently regulated by intra- and inter–molecular interactions. Interaction of the cis-inhibitory cysteine-histidine (CH) domain with the RecA2 domain maintains UPF1 in a state of low catalytic activity (Fig. 1A). This inhibition is relieved upon binding of the core NMD factor UPF2 to the CH domain, which induces a large conformational change and switches the helicase from an RNA-clamping state to an RNA-unwinding state[24,25]. Binding of UPF2 also promotes UPF1 phosphorylation by the PI3K-like kinase SMG1 and triggers release of RNA from UPF1[26–28]. As such, UPF2 is not essential for NMD but can modulate it by influencing UPF1-RNA association kinetics and through protein–protein interactions mediated via the exon junction complex (EJC), which recruits the NMD machinery to defective mRNAs[29–31].

While the function of UPF1 has been extensively studied in NMD, its role in the decay of functional mRNA is less understood from a mechanistic point of view. In this study, we describe molecular mechanisms for the recruitment, function, and activation of UPF1 in RD histone mRNA decay. Using single-particle cryogenic electron microscopy (cryoEM), we determined the structure of human UPF1 bound to the histone SL, which sheds light on the mechanism of remodeling of the SL by the helicase. Our biochemical analyses elucidate how the unwinding activity of UPF1 is finely regulated via protein–protein interactions with SLBP and UPF2, as well as its impact on SL degradation by 3′hExo and the overall stability of histone mRNA in cells. This study shows, for the first time, a direct molecular link between RNA unwinding by UPF1 and RNA degradation in cells and reveals an intricate interplay among key players in the pathway to ensure efficient and timely degradation of RD histone mRNA.

## Results

### UPF1 unwinds the histone SL and facilitates its degradation by 3′hExo

Previous studies by Kaygun and co-workers established a requirement for UPF1 in RD-histone mRNA decay[8]. To dissect the molecular mechanisms of UPF1 in histone mRNA decay, we first investigated the ability of UPF1 to unwind the histone SL. We employed a fluorescence-based nucleic acid unwinding assay, where a short fluorescence-labeled DNA strand hybridized to the 3′ end of an RNA serves as the substrate[32]. The histone SL was incorporated in the RNA strand, upstream of the complementary DNA annealing site at the 3′ end (Fig. 1B, SL RNA). A substrate containing an unstructured RNA annealed to the same fluorescent DNA strand was used as a control (Fig. 1B, linear RNA). ATP-dependent displacement of the DNA strand from the RNA:DNA hybrid by the helicase is monitored over time. The constitutively active UPF1 variant lacking the inhibitory CH domain (UPF1-Hel, Fig. 1A) was used in these experiments. As previously reported, UPF1-Hel unwound >75% of the unstructured linear RNA within the first 10 min. We observed a similar efficient unwinding of the SL RNA substrate by UPF1-Hel, albeit at a slower initial rate compared to the linear RNA substrate (Fig. 1B).

We next set out to address the mechanism of degradation of the histone SL by 3′hExo and the role of UPF1 therein. 3′hExo is a member of the DEDDh family of exonucleases[33]. Atypical of this family, 3′hExo also contains an N-terminal SAP (SAF-box, Acinus and PIAS) domain. The 26 nucleotide (nt) mature SL generated upon 3′ end processing of the histone pre-mRNA is trimmed by 2–3 nts in the cytoplasm by 3′hExo. If shortened past 3 nts, it can be oligouridylated by the terminal uridyl transferase, TUT7[34–37]. The two opposing reactions maintain the SL with a 3 nt extension (at a length of ~24 nt) in the cytoplasm over the S-phase of the cell cycle. With the onset of histone mRNA decay at the end of the S-phase or when DNA replication is inhibited, 3′hExo degrades the SL RNA in a distributive manner, leading to accumulation of distinct SL RNA intermediates[16,38]. First, we quantitatively determined the binding affinity of 3′hExo to the 26-nt SL RNA (SL26), the cytoplasmic variant (SL24) and a stable degradation intermediate (SL19) by fluorescence anisotropy (Supplementary Fig. 1A). SL19 lacks 7 nt from the 3′ end and was identified by End-Seq analysis of S-phase cells when DNA synthesis was inhibited[16,34,39]. 3′hExo binds the SL19 RNA with an affinity comparable to that for SL26 and SL24, suggesting that accumulation of stable intermediates is not due to dissociation of 3′hExo from the SL RNA upon weaker binding to the degradation intermediates.

To analyze the interplay of 3′hExo and UPF1 in histone SL degradation, we adopted a biochemical approach where we compared the 3′−5′ degradation of an SL RNA substrate by 3′hExo in the absence and presence of UPF1. The RNA substrate (60N-SL19) used in this experiment contains a shortened 3′ stem with only four base pairs (corresponding to SL19) and a 60-nt 5′ overhang corresponding to the 3′-UTR stretch preceding the SL in H2bc (Supplementary Fig. 1B). To prevent heterogeneity of the 3′ end length/sequence, we produced this substrate as an HDV ribozyme-fusion by in vitro transcription. Auto-cleavage of the precursor RNA by HDV ribozyme releases 60N-SL19, ensuring identical 3′ ends in all substrate molecules (Supplementary Fig. 1B). Our initial studies showed that 60N-SL19 is more amenable to degradation by 3′hExo than 60N-SL24 (Supplementary Fig. 1C). We reconstituted the substrate with 3′hExo (and UPF1-Hel, where indicated) and added ATP/MgCl$_2$ to initiate the reaction. In absence of UPF1, we observed degradation starting from 10 min after initiation of the reaction, and a steady progress of the reaction up to 60 min. Although the initial substrate was significantly depleted by 60 min, it was not completely degraded, evident from the accumulation of stable intermediates (Fig. 1C, left panel, lanes 7–12). Addition of UPF1-Hel increased the rate of the reaction but did not change the intermediates generated. Accumulation of the first intermediate was observed as early as 10 min and progressed beyond the intermediate obtained in absence of UPF1 at 20 min (Fig. 1C, left panel, lanes 13–18). UPF1 enhanced the efficiency of degradation by 3′hExo by as much as 30–40% in the later time points of the reaction (Fig. 1C, right panel). Interestingly, an identical pattern of degradation was observed upon addition of higher amounts of 3′hExo (Fig. 1C, lanes 1–7), suggesting that UPF1 facilitates degradation of the histone SL but is not necessary for enabling it, consistent with 3′hExo being a distributive enzyme.

## Table 1 | Cryo-EM data collection, refinement, and validation statistics

| Data collection and processing | |
|---|---|
| Microscope | FEI Titan Krios G3i |
| Voltage (keV) | 300 |
| Camera | Falcon 3EC |
| Magnification (nominal/calibrated) | 96,000 |
| Pixel size at detector (Å/pixel) | 0.819 |
| Total electron exposure (e⁻/Å²) | 44 |
| Exposure rate (e⁻/pixel/s) | 0.7 |
| No. of frames collected during exposure | 33 |
| Defocus range (µm) | 0.80–2 |
| Automation software | EPU 3.8.1 |
| Micrographs collected (no.) | 5498 |
| Micrographs used (no.) | 5074 |
| Total extracted particles (no.) | 3,102,505 |
| Final particles (no.) | 300,222 |
| Point-group or helical symmetry parameters | C1 |
| Resolution (global, Å) | |
| FSC 0.143 (unmasked/masked) | 4.0/3.6 |
| Resolution range (local, Å) | 2.8–41.00 |
| Map sharpening *B* factor (Å²)/(*B* factor range) | −207 |
| Map sharpening methods | local B-factor |
| Refinement package | PHENIX (1.21.2_5419) real.space.refine |
| **Model composition** | |
| Non-hydrogen atoms | 6825 |
| protein residues | 798 |
| RNA nucleotides | 25 |
| Zn²⁺ ions | 3 |
| **Model refinement** | |
| **Model-map scores** | |
| - CC (mask) | 0.86 |
| - CC (volume) | 0.85 |
| **Average B factors (Å²)** | |
| overall | 199 |
| protein residues | 191 |
| RNA nucleotides | 285 |
| Zn²⁺ ions | 291 |
| **r.m.s.d. from ideal values** | |
| Bond lengths (Å) | 0.002 |
| Bond angles (°) | 0.524 |
| **Validation** | |
| MolProbity score | 2.07 |
| CaBLAM outliers (%) | 2.4 |
| Clashscore | 9.6 |
| Poor rotamers (%) | 2.1 |
| Cβ deviations | 0 |
| EMRinger score | 0.66 |
| **Ramachandran plot** | |
| Favored (%) | 95.4 |
| Allowed (%) | 4.6 |
| Outliers (%) | 0.0 |
| **Ramachandran plot Z-score, (r.m.s.d.)** | |
| whole | 0.95 (0.31) |

## Table 1 (continued) | Cryo-EM data collection, refinement, and validation statistics

| Data collection and processing | |
|---|---|
| helix | 1.82 (0.32) |
| sheet | 0.46 (0.54) |
| loop | −0.10 (0.31) |

### Recognition of the histone SL by UPF1

To gain insights into the recognition of the histone SL by UPF1 at a molecular level, we determined the cryoEM structure of a UPF1 variant encompassing the CH and the helicase core domains (UPF1CHh, Fig. 1A, Table 1) bound to an intact histone SL with an overhang of 12 uridines at its 5′ end (U-SL, Fig. 1D). The uridines prevent the formation of RNA secondary structure in this region and ensure that UPF1 can bind this stretch of RNA as described in previous studies[24,40]. The final dataset of 300,000 particles was used to generate a reconstruction at a global nominal resolution of 3.6 Å, locally extending to 2.7 Å (Supplementary Fig. 1D–I). We observed ordered density for the UPF1 CH domain, the helicase core (RecA1 and RecA2 domains) and the auxiliary domains 1B and 1C, allowing us to unambiguously place known atomic models of the individual domains and rebuild them locally (Fig. 1D and Supplementary Fig. 1J). Well-defined density, corresponding to single-stranded RNA, was visible in the RNA-binding channel of the UPF1 helicase core (Fig. 1D). Additional density at lower local resolution was observed between domains 1B and 1C, adjacent to the 3′ end of the single-stranded RNA (Fig. 1D and Supplementary Fig. 1I). 4 uridine nts of the 12U stretch and 21 nts of SL26 (C1-C21) were modeled into this density (Fig. 1E and Supplementary Fig. 1K). The overall architecture of UPF1-CHh bound to the U-SL RNA is very similar to that of the X-ray crystal structure of yUpf1 in its transition state[24] (bound to 15U RNA and ADP:AlF₄⁻) and to the cryoEM structure of human UPF1 bound to 15U RNA and AMPPNP[40], with only slight differences in the relative orientation of the RecA domains and domain 1B that we attribute to the absence of ATP in our structure (Supplementary Fig. 2A). UPF1 adopts a closed conformation where the CH domain clamps onto the RecA2 domain, which is typical of the RNA-bound state. The single-stranded RNA threads through a channel formed by the CH and RecA2 domains on one side and domains 1B, 1C and RecA1 on the other, while the SL is positioned at the entry of the channel. In addition to the 4 uridine nts, 5 nts from SL26 that are upstream of the stem are part of the single-stranded RNA stretch and are held in place by a conserved network of interactions observed in other UPF1-RNA structures. The stem of the RNA does not interact with UPF1 but appears to be sterically positioned by domain 1C (Fig. 1E).

### Binding of UPF1 distorts the histone SL to facilitate its degradation

Comparison of the structures of UPF1-bound SL RNA and the SL RNA in complex with SLBP and 3′hExo (PDB-ID 4L8R) revealed interesting features of the RNA. While the conformation of nucleotides C8 to C21 in both structures is very similar, that of their 5′ flanking sequences, including the first 2 nts of the stem, deviates considerably. Instead of a sharp turn between A5 and G6 that changes the direction of the RNA in SLBP-bound SL, nucleotides 5′ of G7 in UPF1-bound SL extend into the RNA-binding channel of UPF1 (Fig. 2A). Nucleotides flanking C21 on the 3′ side are disordered in the UPF1-bound SL structure (Fig. 2A and Supplementary Fig. 2B). NMR spectroscopy studies showed that the 5′ and 3′ flanking sequences of the histone SL alone are highly flexible in solution while the structure of the rigid SL is well-defined[41,42]. Nucleotides G6-U11 form the 5′ arm and A16-C21 form the 3′ arm of the stem, as in the SLBP-bound SL (Supplementary Fig. 2B). Surprisingly, we observed melting of the first G:C base-pair of the SL (G6:C21) and positioning of the unpaired G6 nucleotide in the RNA-binding channel

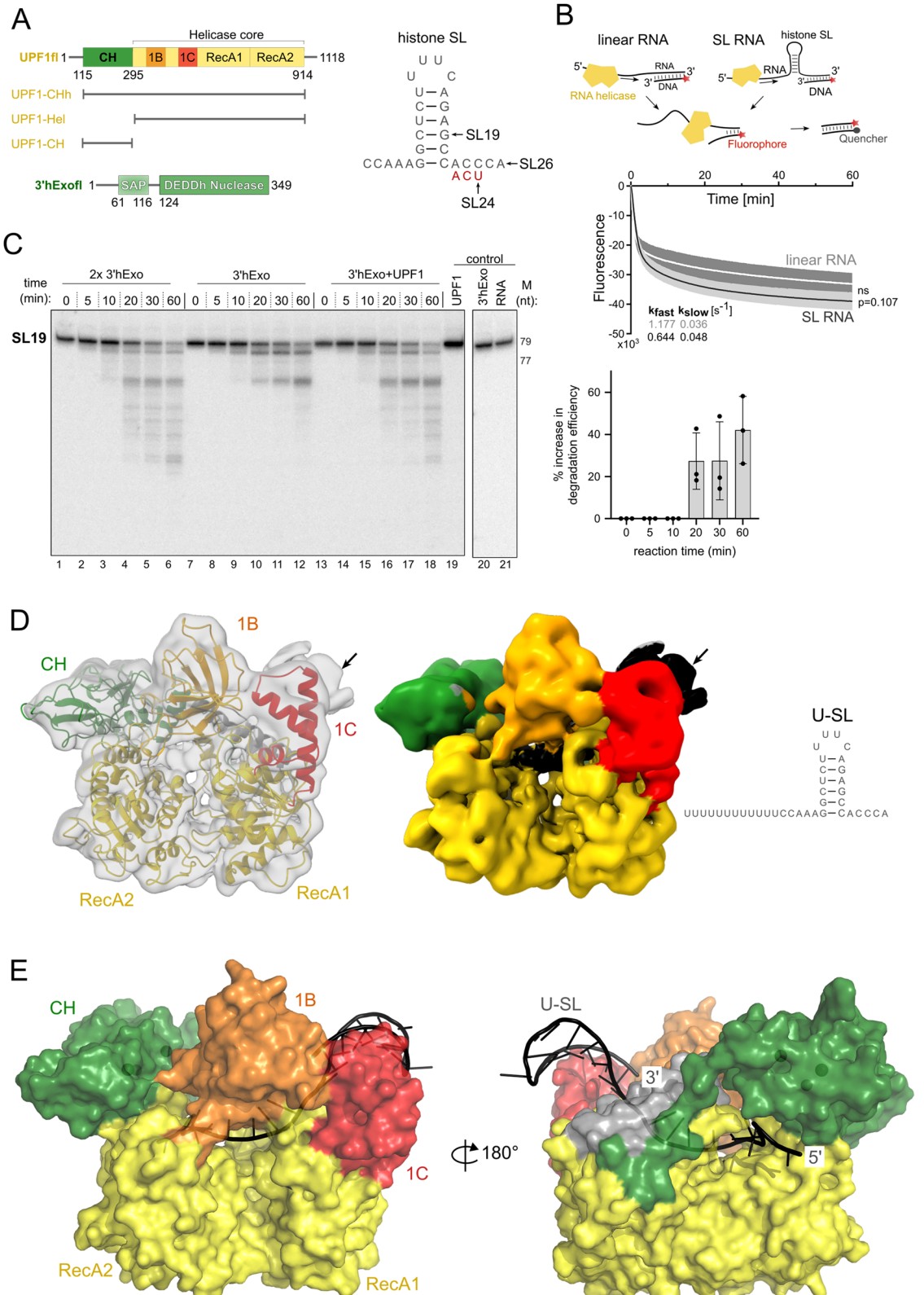

of UPF1 (Fig. 2B). The loop preceding domain 1C of UPF1 is positioned at the base of the stem of the RNA substrate, between the last paired bases (G7:C20) and the first unpaired bases (G6 and C21). A lysine residue within this loop (K547) is wedged between G7 and C20, causing G7 to be distorted out of plane, which in turn weakens the G7:C20 base-pair and disrupts its stacking interaction with C8 (Fig. 2B and Supplementary Fig. 2C). As previously reported by Chapman and co-

workers, mutation of residues in this loop (A546, K547, and S548) significantly impaired ATPase and unwinding activities of the resultant protein, UPF1 AKS-HPA (Supplementary Fig. 2D)[22].

Based on our structural data, we hypothesized that binding of UPF1 to the histone SL is sufficient to facilitate its initial degradation by 3′hExo, without the need for active unwinding. To test this hypothesis, we compared degradation of 60N-SL19 RNA by 3′hExo in the presence

**Fig. 1 | Functional and structural investigation of the interplay of UPF1 with the histone SL. A** Domain organization of human UPF1 and 3′hExo. UPF1 variants used in this study are shown below the schematic. The sequence of the histone SL and variants are indicated. **B** Unwinding activity of UPF1-Hel on substrates containing or lacking the histone SL (SL and linear RNA, respectively). Top: experimental setup of the nucleic acid-unwinding assay. Bottom: Time-dependent measurements of UPF1-Hel unwinding activity on the linear (white) and SL RNA (black) substrates. Data represent the mean of 3 independent experiments, with technical duplicates for every experiment. Shaded areas represent the standard deviation. An unpaired two-tailed t-test ($p = 0.107$) indicates that the difference between unwinding of the linear and SL-RNA substrate is not significant (denoted by ns). The data were fitted to a two-state decay model to obtain the first-order rate constants, $k_{fast}$ and $k_{slow}$. **C** Time-dependent analysis of degradation of 60N-SL19 RNA by 3′hExo in the absence and presence of UPF1-Hel. Addition of UPF1 increases the efficiency (rate

and progression) of the degradation reaction, generating a pattern that is comparable to addition of 2x-3′hExo. Densitometric analysis on the urea-PAGE gel is shown in the right panel. The columns and error bars of the plot represent mean values and standard deviation derived from analysis of 3 independent experiments (shown as data points). **D** CryoEM reconstruction of UPF1-CHh:U-SL RNA. Left: density map (gray) obtained from cryoEM analysis with individual domains of UPF1 (colored according to the schematic in **A**) modeled in. Middle: 3D reconstruction colored by domains as in (**A**). The additional density is colored black. Right: Schematic of the U-SL RNA. The SL was modeled into the density indicated by a black arrow. **E** Structural overview of UPF1-CHh bound to the U-SL RNA. UPF1-CHh is shown in a surface representation and the U-SL RNA as a cartoon. The 5′ single stranded stretch of the RNA threads into the UPF1-RNA binding channel while the SL is positioned at the entry site of the channel formed by domains 1B, 1C and the stalk helices (shown in gray).

of wildtype (wt) UPF1-CHh and a catalytically inactive variant, UPF1-K498A (Fig. 2C). Both wt UPF1 and UPF1-K498A enhance the rate of SL degradation by 3′hExo, although the observed effect is slightly weaker with UPF1-K498A (Fig. 2C, left panel: compare lanes 8–10 with 14–16; right panel: compare 60 min time-points). These observations strongly suggest that the increase in the rate of decay upon addition of UPF1 is not entirely due to ATP-dependent unwinding of the SL but rather results from an effect induced by binding of UPF1 to the SL. We propose that partial melting of the histone SL upon binding of UPF1 makes it more amenable to degradation by 3′hExo in cells. ATP-dependent unwinding of the SL by UPF1 further enhances the rate of histone mRNA decay[8].

## UPF1 is recruited to the histone mRNP by direct protein–protein interactions with SLBP

The molecular mechanisms of recognition of the histone SL by UPF1 raise several questions about recruitment of the helicase. The precise positioning of UPF1 on the histone mRNA observed in the CLIP experiments, upstream of the SLBP-binding site suggests a mechanism of active recruitment rather than scanning of the 3′-UTR by UPF1. Kaygun and co-workers previously showed that UPF1 co-immunoprecipitates with SLBP within minutes of inhibition of DNA replication[8]. The interaction is independent of RNA, suggesting that the association of UPF1 with the histone mRNP is assisted by protein–protein interactions. We therefore tested whether UPF1 and SLBP interact in vitro. Three UPF1 proteins systematically encompassing or lacking the CH and helicase core domains, UPF1-CHh, UPF1-Hel, and UPF1-CH, were tested for their ability to bind SLBP in GST-pulldown assays. GST-UPF1-CHh and UPF1-Hel showed a strong interaction with full-length SLBP (SLBPfl), suggesting that UPF1 engages SLBP via direct protein–protein interactions mediated by its helicase core (Fig. 3A, lanes 1 and 2), independent of SLBP binding to the SL RNA. GST-UPF1-CH showed no binding to SLBP indicating that the CH domain does not participate in this interaction (Fig. 3A, lane 3). We next proceeded to map the UPF1-binding region on SLBP by GST-pulldown. SLBP contains a central RNA-binding domain (RBD), flanked by intrinsically disordered regions (IDRs) at the N- and C-termini. We tested four SLBP proteins for their ability to bind UPF1: SLBPfl, SLBP-N that contains the N-terminal IDR and the RBD, SLBP-C encompassing the RBD and the C-terminal IDR and the SLBP-RBD alone (Fig. 3B). GST-UPF1-Hel was used as a bait in this GST pulldown experiment. We found that GST-UPF1-Hel interacted with SLBPfl and SLBP-N but not with SLBP-RBD or SLBP-C, suggesting that the primary binding site for UPF1 resides in the N-terminal IDR of SLBP (Fig. 3B and Supplementary Fig. 3A). SLBPfl and SLBP-N (but not SLBP-C) migrated as doublets on SDS-PAGE gels and were converted to single faster-migrating species upon treatment with calf intestinal phosphatase (CIP) (Supplementary Fig. 3B), indicating partial phosphorylation of SLBP, likely on T60-T61 within the N-terminal IDR[43] As both the phosphorylated and

unphosphorylated SLBP proteins were co-precipitated with UPF1, it appears that phosphorylation does not impact the interaction of SLBP with UPF1. Likewise, UPF1 phosphorylation also does not affect its binding to SLBP as UPF1fl expressed in HEK 293 efficiently co-precipitates SLBP and no change in binding was observed upon phosphatase treatment of the cell lysate (Supplementary Fig. 3C). This is consistent with previous observations where phosphorylation of UPF1 was shown to be restricted to the N- and C-terminal IDRs flanking its helicase core[44]. Our observations define the direct protein–protein interactions between UPF1 helicase core and the SLBP N-terminal IDR and are in line with the previously reported interactions between these factors[15].

Since the substrate for histone mRNA degradation is the SLBP-bound mRNA, we asked whether binding of SLBP to the SL RNA influences its binding to UPF1. To this end, we performed analytical size-exclusion chromatography (SEC) to assess the interaction between UPF1-Hel and SLBP-N in the absence and presence of the SL26 RNA. A stable complex between UPF1 and SLBP was formed only upon addition of SL26 RNA, as indicated by the lower retention volume of peak 1 and the presence of all three components in lane 1 of the SDS- and urea-PAGE gels (Fig. 3C and Supplementary Fig. 3D). SL26 RNA is necessary for SLBP to fold correctly and adopt a functional conformation as SLBP-N shows a propensity for aggregation in the absence of SL RNA (Fig. 3C, peak 4). We conclude that although the interaction between SLBP and UPF1 is driven by direct protein–protein interactions, the histone SL plays a key stabilizing role, likely by maintaining SLBP in a functionally competent conformation since the RNA binding domain is not stably folded in the absence of RNA.

## The N-terminal IDR of SLBP engages UPF1 to mediate efficient histone mRNA decay

To gain further insights into the UPF1-SLBP interaction in the context of the mRNP, we performed crosslinking mass-spectrometry (CXMS) on an in vitro reconstituted ternary complex of UPF1-Hel, SLBP-N, and the U-SL RNA, using the long-range crosslinker BS3 (bis(sulphosuccinimidyl)-suberate) that can span lysine residues 30 Å apart (Supplementary Fig. 3E, F, bottom left panel). Consistent with the position of the SL with respect to the UPF1 helicase core, we found multiple crosslinks between the SLBP N-terminal IDR and the RecA1 and 1B domains of UPF1 (Supplementary Table 2, Fig. 3D and Supplementary Fig. 3F). Fewer crosslinks were found to the RecA2 domain. Although the SLBP-RBD did not bind UPF1 in our biochemical experiments, we observed cross-links between 2 of the 10 lysine residues within SLBP-RBD to UPF1 (Supplementary Fig. 3F, bottom right panel). It is likely that the SLBP-RBD bound to the SL RNA is positioned in proximity to the UPF1 helicase core, without a direct interaction, and is therefore crosslinked. Correspondingly, lysine residues of SLBP that directly contact the SL RNA were not crosslinked to UPF1, validating the

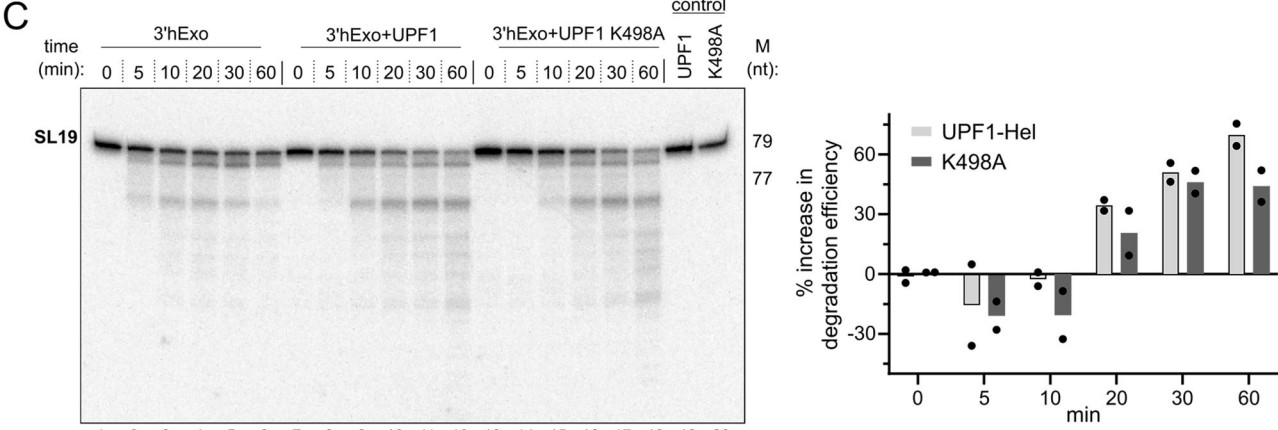

**Fig. 2 | Binding of UPF1 distorts the histone SL to facilitate its degradation.**
**A** Superposition of the UPF1-bound SL RNA (black) with the SLBP-bound SL RNA (cyan). SLBP has been omitted for clarity. Binding of UPF1 alters the conformation of the 5' arm of the SL (nucleotides G6 and G7) and the 5' flanking sequence (see inset), which enters the RNA-binding channel of UPF1. **B** View of the interaction of UPF1 with the SL RNA. Domain 1B has been omitted in this view. Unpaired nucleotides of the SL are shown in black, while the 5' and 3' arms of the SL are shown in light and dark blue, respectively. A schematic of the 26 nt histone SL (SL 26) is shown as a reference. Binding of UPF1 completely disrupts the G6:C21 base-pairing

in SL26 and distorts the conformation of G7, weakening base-pairing and stacking interactions with C20 and C8, respectively (refer to Supplementary Fig. 2B). The loop containing residues A546, K547, and S548 of UPF1 is positioned at the base of the SL stem, wedging the sidechain of K547 in between the two arms of the stem to splay them apart (see also Supplementary Fig. 2C). **C** Time-dependent analysis of degradation of 60N-SL19 RNA by 3'hExo in presence of UPF1-Hel (lanes 7–12) and the catalytic mutant, UPF1-K498A (lanes 13–18). The inactive UPF1 variant also facilitates degradation of the SL by 3'hExo. Densitometric analysis and quantification of the degradation reactions (right) were done as described for Fig. 1C.

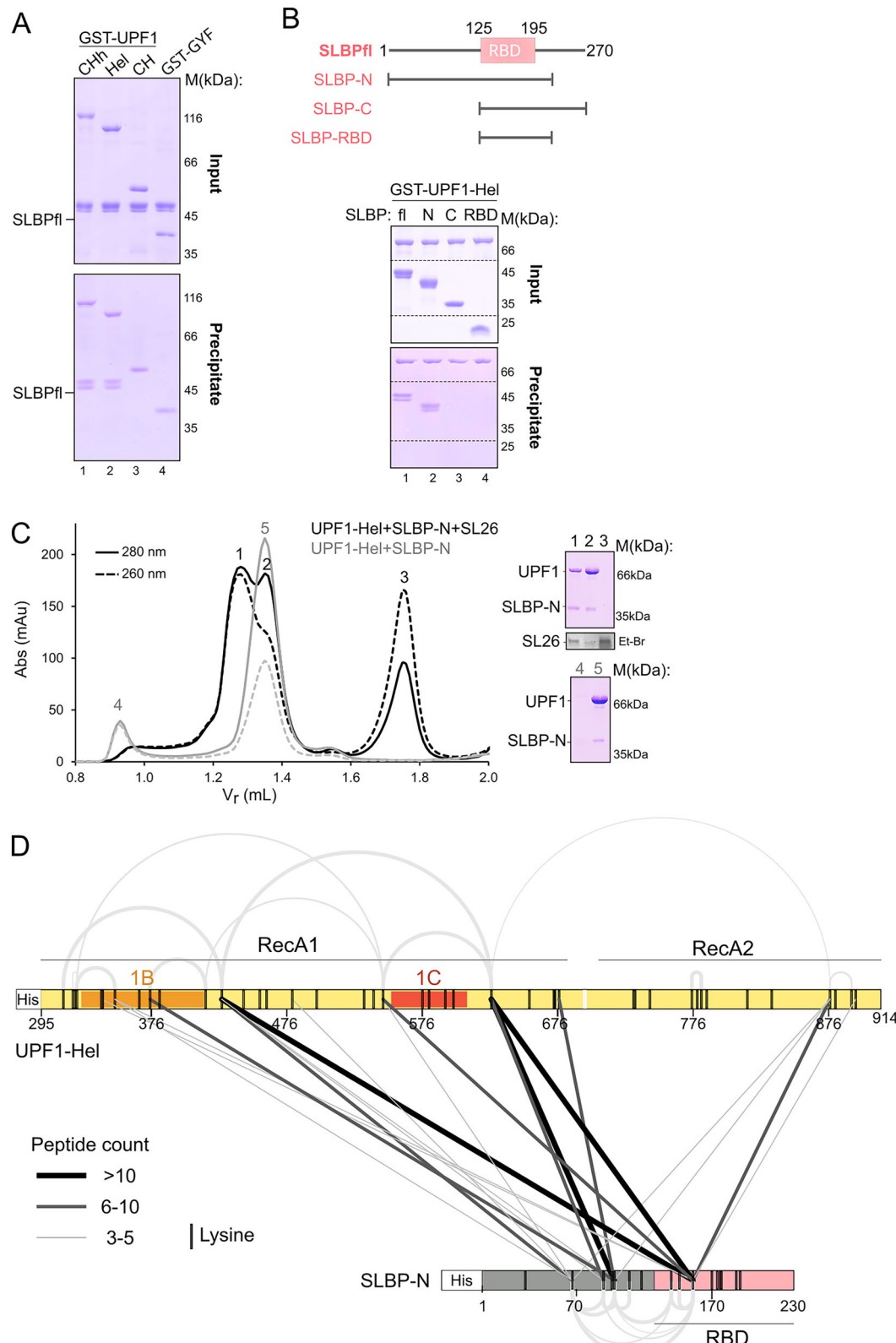

specificity of the approach in mapping UPF1–SLBP interactions in the context of the histone decay mRNP.

We next proceeded to narrow down the UPF1-binding site within the N-terminal IDR of SLBP. To this end, we generated a series of SLBP N-terminal truncation variants and analyzed their binding to GST-UPF1-Hel using GST-pulldowns (Fig. 4A). We observed a weakening of the SLBP-UPF1 interaction upon deletion of the first 30

amino acids, with a complete loss of binding upon removal of the first 60 amino acids of SLBP. Interestingly, internal deletion of stretches of approximately 30 residues from the N-terminal IDR (amino acids 21–58 or 31–60), which are highly conserved in vertebrates, also weakened the affinity of SLBP for UPF1 (Fig. 4A and Supplementary Fig. 4). Taken together, our results suggest that the UPF1-interacting motif of SLBP resides within the first 60 residues of

**Fig. 3 | The N-terminal IDR of SLBP engages the UPF1 helicase core via direct protein–protein interactions. A** Mapping the SLBP-binding site on UPF1. GST-pulldown assays were performed with GST-UPF1 proteins and SLBPfl. GST-GYF serves as a negative control. The top and bottom panels show the inputs and bound fractions (precipitate) in this and all subsequent GST-pulldown experiments. The helicase core of UPF1 is sufficient for its interaction with SLBP. All GST-pulldown assays in this study were performed independently three times with similar results. **B** Mapping the binding site for UPF1 on SLBP. Top: schematic representation of the domain arrangement of human SLBP and the SLBP variants tested. Bottom: GST-pulldown using GST-UPF1-Hel and SLBP proteins. The N-terminal IDR of SLBP is necessary for binding to UPF1. A complete gel including negative controls with GST-GYF is shown in Supplementary Fig. 3A. **C** Analytical size exclusion chromatography (SEC) of UPF1-Hel and SLBP-N in the absence (gray traces) and presence (black traces) of SL26 RNA. PAGE analyses corresponding to the SEC runs are shown on the right (top: with SL26, bottom: without SL26). The SL RNA was visualized by

ethidium bromide (Et-Br) staining of the urea-PAGE gel. Binding of SL26 RNA to SLBP-N stabilizes the protein. The experiment was performed twice with similar results. **D** Linkage map showing the interlinks between UPF1 and SLBP, derived from crosslinking-mass spectrometry analysis of the UPF1-Hel:SLBP-N:U-SL complex (see Supplementary Fig. 3E) using BS3. The analysis was performed in triplicate on the same sample. Interlinks are depicted as black or gray lines, where linewidth corresponds to the count of peptides identified for each crosslink. Intralinks within each protein are shown as gray arcs, represented according to peptide count, as for the interlinks. The schematics represent the protein variants used for reconstitution of the complex and are colored according to Figs. 1A and 3B. Positions of all lysine residues in each protein are indicated by black vertical lines in the schematic. See also Supplementary Fig. 3F for mapping of UPF1-SLBP crosslinked residues onto the UPF1:U-SL RNA and SLBP:SL26 RNA structures. Mass spectrometry data are presented in Supplementary Table 2.

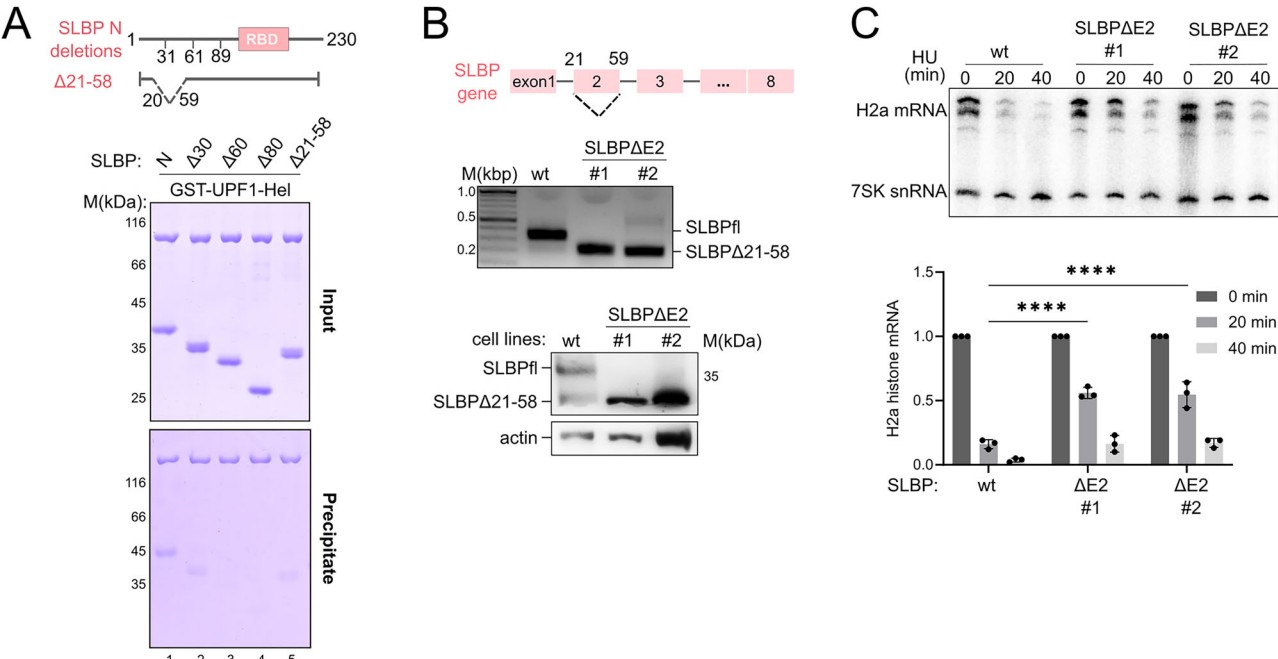

**Fig. 4 | Disruption of the UPF1-SLBP interaction reduces the rate of RD histone mRNA decay in cells. A** GST-pulldown assay of GST-UPF1-Hel and truncations of SLBP-N. The positions of the N-terminal truncations as well as the deletion within the N-IDR, are depicted in the schematic above. Removal of the first 30 residues weakened its binding to UPF1, while removing 60 or 80 residues abolished binding. Deleting amino acids 21–58 [corresponding to exon 2 of the SLBP gene] reduced binding more than deleting the first 30 amino acids (see also Supplementary Fig. 4A). A complete gel including negative controls with GST-GYF is shown in Supplementary Fig. 4B. **B** Top: Gene organization of the human SLBP gene. Deletion of the 111 nt exon 2 of the endogenous SLBP gene by CRISPR/Cas results in an in-frame deletion removing residues 21–58 in the SLBP protein. Middle: PCR analysis of the genome locus of the SLBP gene in wild-type and CRISPR/Cas edited HCT116 cells to confirm successful deletion of exon 2 in the endogenous SLBP gene.

Bottom: Western blot analysis of wildtype and edited cell lines (SLBPΔE2) using an SLBP-specific antibody shows expression of wildtype and truncated SLBP (Δ21–58). Actin (lower panel) was used as a loading control. **C** Weakening the UPF1–SLBP interaction significantly reduces the efficiency of histone mRNA decay. HCT116 cells containing the SLBP deletion in both alleles were treated with 5 mM hydroxyurea for the indicated time. Total RNA was prepared and analyzed for histone H2a mRNAs by Northen Blotting, using 7SK RNA as an internal control. The data are quantified in the bottom panel. Quantitative data were derived from Phosphor-Imager analysis of the Northern blot and are a representation of the mean values and standard deviation from 3 independent experiments. An unpaired two-tailed t-test was performed to test for significances of the differences observed (adjusted p-value < 0.0001, denoted by ****).

its N-terminal IDR and facilitates recruitment of UPF1 to the 3′-UTR of histone mRNA, proximal to the histone SL.

We investigated the effect of binding of UPF1 and SLBP on histone mRNA decay in cells by taking advantage of the fact that the second exon of the SLBP gene (111 nt encoding amino acids 21–58, Fig. 4B) could be removed to create an in-frame deletion. This deletion was engineered in HCT116 cells using CRISPR/Cas to produce two clonal cell lines (SLBPΔE2). The SLBPΔE2 cell lines exclusively express the SLBPΔ21–58 protein at levels comparable to full-length SLBP (Fig. 4B), and the cells grew normally, indicating that this region is not essential. To determine if deletion of this region affects

histone mRNA degradation, wild type (wt) and SLBPΔE2 cells were treated with hydroxyurea, an inhibitor of DNA replication, to induce histone mRNA decay and the levels of histone H2a mRNA were monitored by Northern blotting (Fig. 4C). Wild type cells expressing full-length SLBP showed a rapid reduction in H2a transcript levels within 20 min of hydroxyurea treatment and >90% depletion of the transcript after 40 min. In comparison, H2a mRNA was not rapidly degraded in the SLBPΔE2 cell lines, with as much as 50% of starting amounts remaining after 20 min of induction of histone mRNA decay (Fig. 4C, bottom panel). A further depletion of histone mRNA levels was observed after 40 min, but the amount

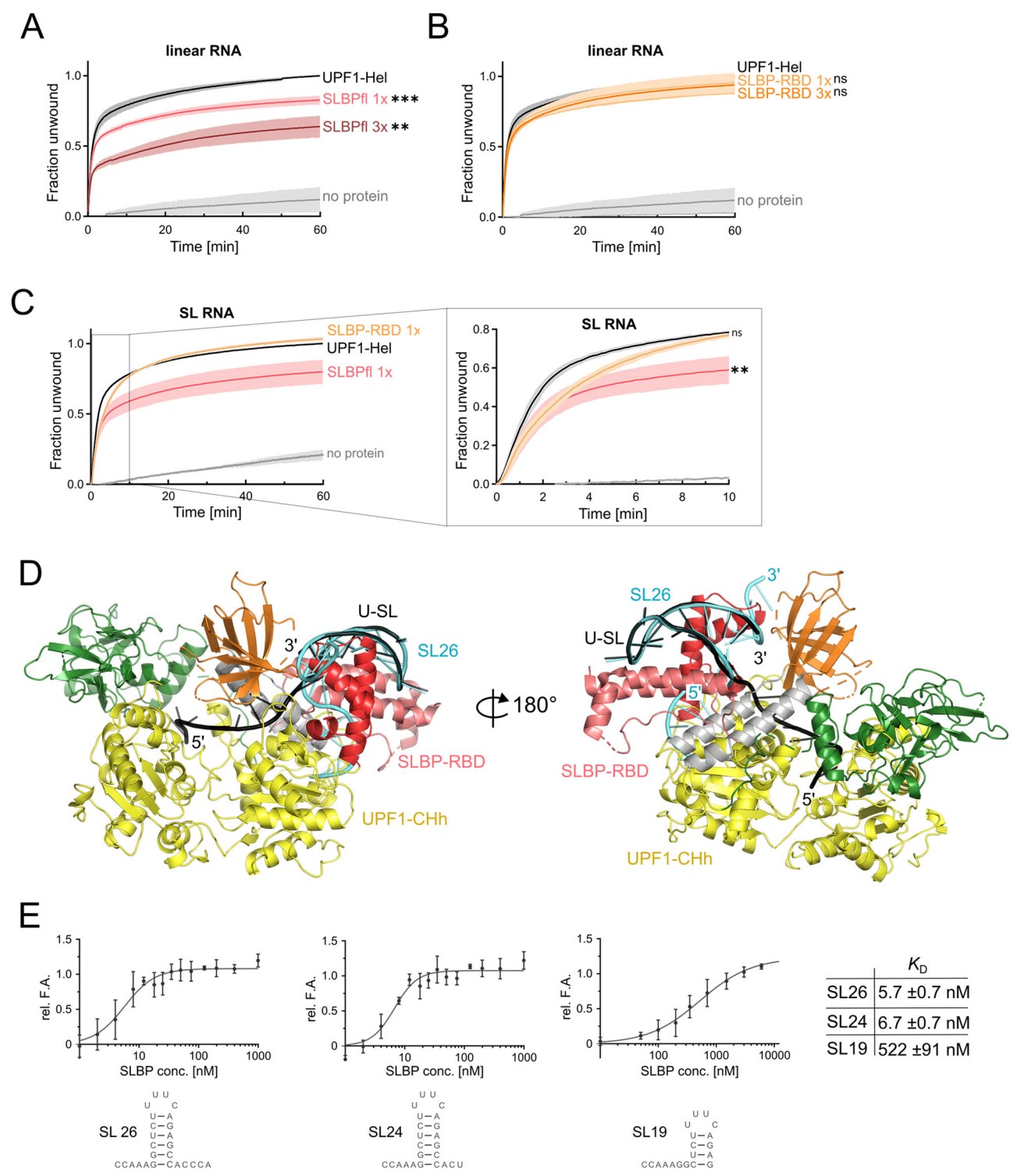

of H2a mRNA remaining in cells expressing SLBPΔ21–58 was higher than that in wild-type cells. We conclude that an interaction between UPF1 and SLBP is crucial for ensuring efficient and rapid degradation of RD histone mRNA when DNA replication is inhibited. It is important to mention that the N-terminal IDR of SLBP also engages other factors, such as those involved in nuclear export and translation of histone mRNA, although these interactions occur at distinct points of a histone transcript's lifetime in the cell[45,46]. It is possible that interactions of the SLBP N-IDR with yet unknown cellular factors might further influence the rate of histone mRNA decay.

## SLBP directly and indirectly impacts unwinding of the histone SL by UPF1

Our structural and biochemical findings present a complex picture of the interplay of UPF1, SLBP, and the histone SL. Given that the catalytic activity of UPF1 is extensively regulated by intra- and inter-molecular protein–protein interactions[24,47], we sought to investigate whether binding of SLBP to UPF1 affects its catalytic activity. We performed fluorescence-based nucleic acid unwinding assays to compare the ability of UPF1 to unwind the linear RNA substrate, which lacks the SL (described above) in the absence and presence of SLBP. The experimental data were fitted to a two-phase decay model with a fast

**Fig. 5 | SLBP restrains unwinding of the histone SL by UPF1 and is displaced from the SL RNA prior to degradation.** Unwinding activity of UPF1-Hel on the linear RNA substrate in presence of SLBPfl (**A**) or SLBP-RBD (**B**). The amounts of SLBP proteins added in each reaction are indicated. The fraction of substrate unwound was plotted against time. Data are presented as in Fig. 1B. SLBP bound to the SL has an inhibitory effect on the unwinding activity of UPF1. Unpaired two-tailed t-tests were performed to determine the significance of differences observed (** = very significant, $p \leq 0.01$; *** = extremely significant, $p \leq 0.001$; ns = not significant, $p > 0.05$). Individual $p$-values as well as first-order rate constants derived from fitting the experimental data to a two-phase decay model are shown in the corresponding Supplementary Figs. (Supplementary Fig. 5A–C). **C** Unwinding activity of UPF1-Hel on the SL RNA substrate in the presence of equimolar amounts of SLBPfl (pink trace) or SLBP-RBD (orange trace). The inset shows an enlarged view of the first 10 min of the reaction. SLBP-RBD retards unwinding of the SL RNA

substrate by UPF1 in the early stages of the reaction, while SLBPfl reduces the activity of UPF1 throughout the reaction. Data presentation and statistical analysis are as described above (**A**, **B**). **D** Superposition of the UPF1-bound SL RNA (black) with the SLBP-bound SL RNA (cyan). The conformation of UPF1-bound SL RNA is incompatible with tight binding of SLBP (see also Supplementary Fig. 5E, F). Steric clashes of SLBP with domain 1C and a stalk helix of UPF1 suggest that SLBP must be displaced from the histone mRNP as UPF1 unwinds the SL. **E** Quantitative measurement of the binding affinities of SLBPfl for the histone SL and SL degradation intermediate. Data points and error bars are presented as the mean and standard deviation of 3 independent experiments. The dissociation constants ($K_D$) were derived as described in Supplementary Fig. 1A. The binding affinity of SLBP for the degradation intermediate (SL19) is 100-fold lower than that for an SL with an intact stem (SL26/SL24).

unwinding phase(prominent in the early time-points), followed by a slow unwinding phase that is predominant in the later time points (Supplementary Fig. 5A–C). Addition of an equimolar amount of SLBPfl significantly reduced the extent of linear RNA substrate unwound by UPF1-Hel to 66% (Fig. 5A, compare black and pink traces). Increasing the amount of SLBPfl to 3-fold molar excess over UPF1 led to a further decrease in the fraction of substrate unwound (Fig. 5A, brown trace and Supplementary Fig. 5A). Addition of SLBP-RBD which lacks the UPF1-binding region did not result in a decrease in the fraction of substrate unwound by UPF1-Hel, even when added in 3-fold excess to UPF1 (Fig. 5B and Supplementary Fig. 5B). A similar effect was observed with the SL RNA substrate, where addition of SLBPfl, but not SLBP-RBD, led to an overall decrease in the fraction of SL RNA substrate unwound by UPF1-Hel (Fig. 5C and Supplementary Fig. 5C). To ascertain that the observed decrease in unwinding is due to direct UPF1-SLBP interactions, we proceeded to test the effect of the SLBP N-terminal IDR on UPF1 unwinding activity. As this region cannot be stably expressed on its own, we generated a chimeric protein where the SLBP N-IDR is covalently linked to the UPF1 helicase core. The chimeric protein, UPF1-SLBP fusion, showed a strong decrease in unwinding activity relative to UPF1-Hel, indicating that binding of the SLBP N-terminal IDR to the UPF1 helicase core impedes its unwinding activity (Supplementary Fig. 5D).

Interestingly, addition of SLBP-RBD appears to specifically decrease the rate of initial and fast unwinding of the SL-substrate by UPF1-Hel but does not affect the rate of the later slow unwinding phase or the extent of SL-substrate unwound overall (Fig. 5C, inset and Supplementary Fig. 5C). This effect is not observed with the linear RNA substrate, suggesting that it is specific for binding of SLBP to the histone SL. We postulate that when bound to the SL RNA, SLBP creates a roadblock for translocation and must be actively displaced by UPF1 to reach the 3′ end and unwind the RNA:DNA hybrid. Accordingly, the addition of SLBP-RBD only slows down the initial unwinding of the SL RNA substrate but does not affect the fraction of substrate that has been unwound by UPF1-Hel at later time-points. Taken together, our data suggest that SLBP uses two distinct mechanisms to modulate unwinding of the histone SL by UPF1: directly, by binding the UPF1 helicase core, where it reduces the unwinding rate, and indirectly, via the strong association of the RBD with the histone SL, which slows UPF1 progression.

Superposition of the structures of UPF1-bound SL and SLBP-bound SL reveals that the conformation of UPF1-bound SL is incompatible with the binding of SLBP (Fig. 5D). SLBP specifically recognizes G7, base-pairing of which to C20 is partially disrupted. Nucleotides A3-A5 in the 5′ flanking sequence that are contacted by SLBP are positioned in the RNA-binding channel of UPF1 (Supplementary Fig. 5E, F, see also Fig. 2A). Our structure therefore represents a post-displacement state where SLBP has been ejected from the histone mRNA, and UPF1 is poised to unwind the SL, in the absence of a roadblock.

Previous studies on SLBP and the histone SL showed that SLBP binds the 26 nt SL RNA with very high affinity[48], rationalizing its role as a roadblock to translocation of UPF1. To glean further insights into how SLBP acts as a roadblock for UPF1 unwinding and its impact on SL RNA degradation, we set out to determine the binding affinity of SLBP to the SL RNA variants SL26, SL24, and SL19, using fluorescence anisotropy (Fig. 5E). Consistent with previous reports, SLBP binds SL26 with a very high affinity ($K_D$ of ~6 nM)[48]. The cytoplasmic SL, SL24, which contains an intact stem, also shows very strong binding ($K_D$ of ~7 nM) to SLBP. In contrast, the affinity of SLBP for the intermediate SL19 is drastically reduced ($K_D$ of ~500 nM), suggesting that once the SL is partially degraded, SLBP can no longer rebind the intermediate. We argue that this is likely an important event as SLBP also acts as a roadblock for degradation by 3′hExo in our in vitro degradation analysis (Supplementary Fig. 5G) and in previous studies[7].

## Function and recruitment of the UPF1-activator, UPF2, in context of the histone mRNP

Our results indicate that rapid initiation of histone mRNA decay would require activation of UPF1 to efficiently displace SLBP and unwind the histone SL. The core NMD factor UPF2 is a known activator of UPF1 that also functions in other pathways of UPF1-mediated mRNA decay. Binding of UPF2 to UPF1 stimulates its catalytic activity by inducing a large conformational change in the helicase[24–26]. We therefore proceeded to biochemically investigate the involvement of UPF2 in histone mRNA decay. As a first step towards understanding the effect of UPF2 on UPF1 activation in the context of the histone SL-RNP, we tested the unwinding activity of UPF1-CHh on the SL RNA substrate. The UPF1-CHh protein has the *cis*-inhibitory CH domain that retards translocation of the helicase on RNA. As expected, the rate of SL RNA unwinding of UPF1-CHh is lower than that of the constitutively active UPF1-Hel protein, with only ~40% of the substrate unwound in the first 10 min by UPF1-CHh in contrast to ~75% by UPF1-Hel (Fig. 6A, compare black and gray traces). We next investigated the effect of UPF2 on unwinding of the SL RNA by UPF1. The domain structure of UPF2 is shown in Fig. 6C. It contains three *m*iddle-of-e*IF4G* (MIF4G) domains and a C-terminal UPF1-binding domain (U1BD)[25,49]. The truncated UPF2s protein spans the domains MIF4G3 and U1BD, which mediate all known interactions of UPF2[25,50–52]. Consistent with previous studies, addition of UPF2s significantly enhances the unwinding activity of UPF1 (Fig. 6A, compare black and blue traces)[24]. To dissect the effects of SLBP on UPF2-activated UPF1, we measured the rate of unwinding of the SL RNA substrate by UPF1-CHh in presence of SLBPfl, with and without addition of UPF2s. As described earlier, SLBP bound to the SL acts a roadblock and reduces the initial rate of unwinding of UPF1-Hel. Addition of SLBPfl also retards the initial rate of unwinding by UPF1-CHh, which is overridden in the presence of UPF2 (Fig. 6B, compare black, red and purple traces). We therefore concluded that UPF2 can activate the UPF1 helicase on the histone SL RNA. Consistent with these

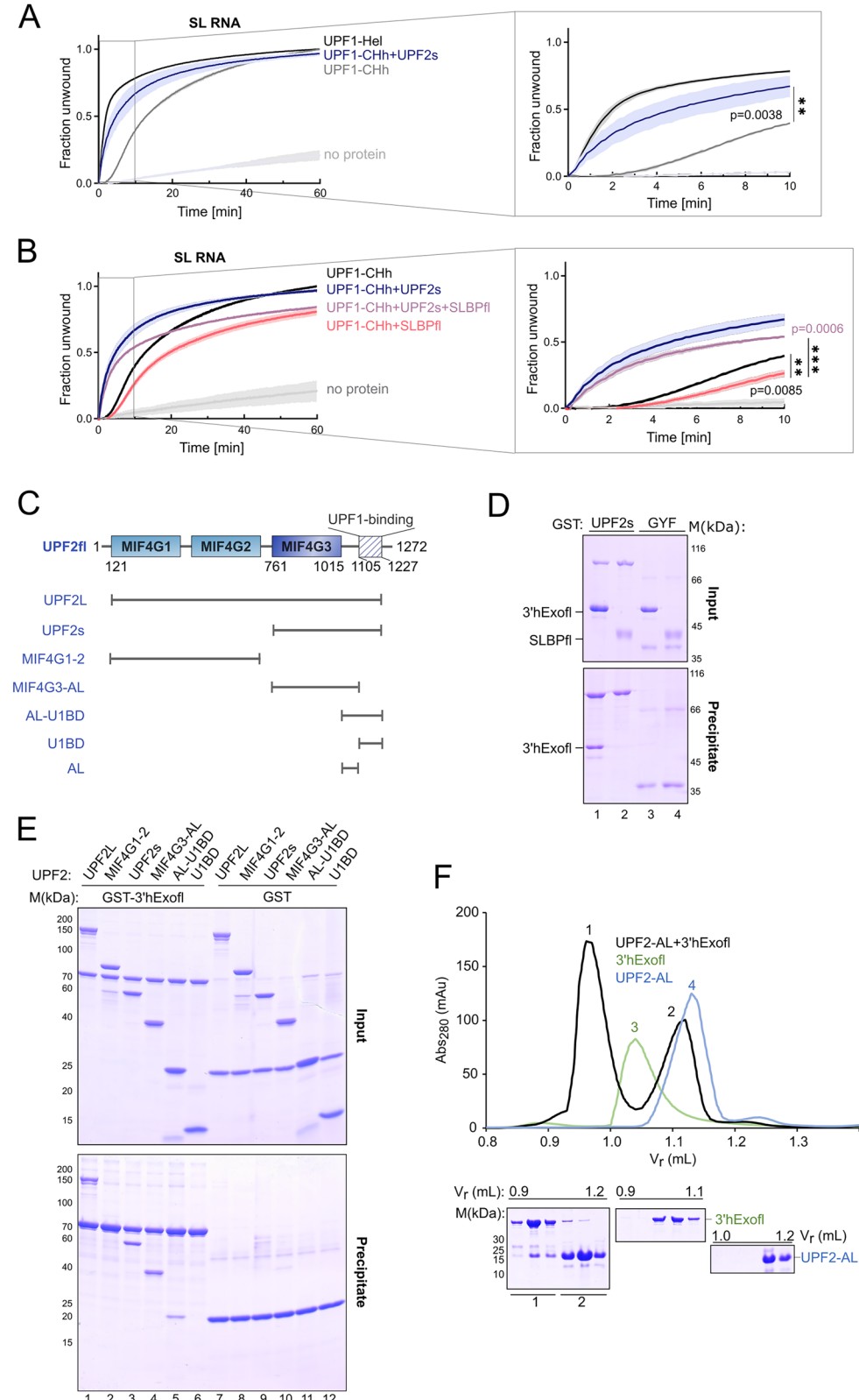

results, we previously showed that a UPF1 protein lacking the CH domain is not active in degrading histone mRNA in vivo[16].

Recent studies show that binding of UPF2 to UPF1 drastically reduces the affinity of the helicase for RNA and that UPF2 cannot remain stably associated with RNA-bound UPF1[28]. It therefore follows that once UPF2 activates UPF1 on the histone mRNA, UPF2 must interact with another protein factor within the mRNP, allowing it to be

in proximity to the helicase and engage it when necessary. To investigate this possibility, we performed GST-pulldowns of GST-UPF2s with SLBP and 3'hExo. UPF2s shows an interaction with 3'hExo but not SLBP (Fig. 6D, lanes 1–2), suggesting that it might be recruited to the histone mRNP via 3'hExo. To map the 3'hExo-binding site on UPF2, we tested interactions of a series of UPF2 constructs with GST-3'hExo (Fig. 6C, E). As expected, we saw an interaction of GST-3'hExo with UPF2L and

**Fig. 6 | The role of the UPF1-activator, UPF2, in context of histone mRNA decay.**
**A** Comparison of the unwinding activity of UPF1-CHh on SL RNA substrate in the absence (gray trace) and presence of UPF2s (blue trace). The unwinding activity of UPF1-Hel is shown as a reference (black trace). Data presentation and statistical analysis for this experiment and Fig. 6B are as described for Fig. 5A, B. The inset shows the first 10 min of the reaction, where the differences in unwinding activity are the largest. The significance of the differences in unwinding observed in this experiment and in Fig. 6B as well as $p$-values are indicated. **B** Unwinding activity of UPF1-CHh on the SL RNA substrate in the presence of either UPF2 (blue trace) or SLBPfl (red trace) or both UPF2 and SLBP (purple trace). The activity of UPF1-CHh in absence of UPF2 and SLBPfl (black trace) has been shown for comparison. The inset shows an enlarged view of the first 10 min of a 60-min reaction. Activation of UPF1 by UPF2 reduces the inhibitory effect of SLBP on the helicase. **C** Schematic representation of the domain organization of human UPF2. The structured middle-of-eIF4G (MIF4G) domains are shown as filled rectangles while the C-terminal U1-binding domain (U1BD) is depicted by a crosshatch. Variants used in this study are shown below. **D** GST-pulldown assay of GST-UPF2s (and GST-GYF as negative control) with full-length SLBP and 3'hExo. 3'hExofl shows a specific interaction with UPF2s while SLBPfl shows no appreciable binding. **E** GST-pulldown assays using GST-3'hExofl as a bait and different UPF2 variants as prey show that all UPF2 proteins encompassing the acidic linker (AL) are co-precipitated by 3'hExo. **F** Analytical SEC of mixture of UPF2-AL and 3'hExo (black trace) confirms that the UPF2-AL is sufficient for formation of a stable complex with 3'hExo. SEC runs of the individual proteins 3'hExofl (green trace) and UPF2-AL (blue trace)1`123 are shown for comparison. The corresponding SDS-PAGE analysis are shown below. See also Supplementary Fig. 6A. The experiment was performed independently three times with similar results.

UPF2s, but not with UPF2-MIF4G1-2, supporting our earlier observation that the binding site for 3'hExo resides in the C-terminal region of UPF2 (Fig. 6E, lanes 1–3). Systematic truncations of the C-terminal region of UPF2 revealed that the U1BD is not involved in mediating interactions with 3'hExo (Fig. 6E, lane 6). The MIF4G3 domain of UPF2 is connected to its U1BD via a stretch of predominantly acidic residues that we refer to as the acidic linker (AL). UPF2 fragments harboring the AL but lacking either the U1BD or the MIF4G3 domains (UPF2 MIF4G3-AL and UPF2 AL-U1BD, respectively) showed strong binding to 3'hExo, suggesting that the 3'hExo-binding site is located within the AL of UPF2 (Fig. 6E, lanes 4 and 5). To corroborate these observations, we reconstituted a complex of 3'hExo with the UPF2-AL by SEC (Fig. 6F). While the AL engaged 3'hExo in a stable complex, a UPF2 variant lacking the AL (UPF2ΔAL) failed to interact with 3'hExo (Supplementary Fig. 6A). We concluded that the AL of UPF2 is necessary and sufficient for mediating a stable interaction with 3'hExo. Reconstitution of a stable ternary complex of 3'hExo-UPF2-UPF1 shows that UPF2 can simultaneously engage with 3'hExo and UPF1 in solution, and that binding of 3'hExo to the UPF2-AL does not perturb the interactions between UPF2-U1BD and UPF1 (Supplementary Fig. 6B).

We next performed GST-pulldowns of 3'hExo truncations with the GST-UPF2s protein. The SAP and nuclease domains co-precipitated with UPF2, suggesting that both domains interact with UPF2-AL (Fig. 7A). To gain insights into the UPF2-3'hExo interaction at a molecular level, we used NMR spectroscopy to monitor the changes in the amide peaks of $^{15}$N-labeled UPF2-AL upon addition of unlabeled 3'hExofl. We observed chemical shift perturbations (CSPs) in the fast exchange regime in the NMR timescale, indicative of relatively weak binding (Fig. 7B). To locate the regions of UPF2-AL most affected in the presence of 3'hExo, we assigned its backbone resonances ($^{13}$C$^{\alpha}$, $^{13}$C$^{\beta}$, $^{1}$H$^{N}$, and $^{15}$N) and plotted the amide CSPs as a function of residue number (Fig. 7C). The largest CSPs were found in a broad region encompassing residues 1030–1090 enriched in glutamate residues, indicating that binding did not occur through a highly localized interface of UPF2. The chemical shifts observed also confirmed that UPF2-AL is intrinsically disordered in solution (Supplementary Fig. 7A) and does not become ordered upon addition of 3'hExo, as shown by the lack of amide dispersion in the bound state. Upon titration of the individual SAP and nuclease subdomains of 3'hExo into $^{15}$N-labeled UPF2-AL, we observed similar regions being perturbed (Supplementary Fig. 7B, C). Therefore, UPF2 interacts with distinct regions of 3'hExo using a similar interface, without making domain-specific interactions. We speculate that a basic groove spanning the SAP and nuclease domains of 3'hExo serves as a docking platform for UPF2-AL, engaging it in low-affinity but high-avidity interactions that allows formation of a stable complex in solution. Using surface plasmon resonance, we determined the dissociation constant ($K_D$) of the 3'hExo-UPF2 interaction to be ~170 μM, in agreement with the $K_D$ of ~200 μM estimated from our NMR titrations (Supplementary Fig. 7D, E).

## Discussion

The studies presented here provide the first mechanistic insights into the molecular interaction network necessary for initiation of regulated histone mRNA decay. It was shown that 3'hExo, when bound to an SL RNA-SLBP complex cannot degrade beyond 2 nts from the 3' end[12]. Identification of distinct decay intermediates resulting from 3'hExo degrading into the SL clearly suggests that initiation of degradation also involves a mechanism that makes the 3' end of the SL accessible to 3'hExo. As the catalytic activity of UPF1 was shown to be important for mediating efficient histone mRNA decay, it followed that the role of UPF1 in histone mRNA decay must be to actively unwind the SL at the 3' end. The cryoEM reconstruction of UPF1 bound to an SL RNA with a 5' overhang (U-SL) reported here shows that binding of UPF1 immediately upstream of the SL distorts its conformation and melts the base of the stem, without ATP binding and hydrolysis. This observation rationalizes our biochemical studies where a catalytically inactive UPF1 variant that is incapable of binding ATP also facilitates decay of the SL RNA by 3'hExo. Our structure indicates that initiation of strand-separation does not require ATP but is likely driven by the energy derived from binding the RNA. ATP-independent unwinding of nucleic acid was initially demonstrated for the Hepatitis C virus helicase, NS3, and the archaeal DNA helicase, Hel308. In Hel308, a β-hairpin at the surface of the RecA2 domain acts as an unwinding element to splay apart the two strands of the DNA[53,54]. We identified a similar unwinding element in UPF1, the AKS-loop in RecA1, which is conserved in UPF1 proteins across species. Melting of the RNA stem by UPF1 bears similarity to local strand-separation of nucleic acids by DEAD-box proteins. However, DEAD-box helicases release the RNA substrate after each cycle of ATP hydrolysis, whereas UPF1 and other translocating helicases use the energy of ATP hydrolysis to processively unwind the RNA substrate[55].

The binding of UPF1 to the histone SL is, in principle, sufficient to unwind the SL and initiate its decay. This implies that UPF1 must be stringently regulated within the histone mRNP, in terms of its access to the SL as well as its catalytic activity, to prevent premature decay. In this study we show that SLBP plays a pivotal role in regulating UPF1 within the histone mRNP by engaging it via direct protein–protein interactions mediated by its N-terminal IDR. The levels of SLBP protein increase by approximately 20-fold during the S-phase of the cell cycle[56]. This ensures that a sufficient amount of SLBP is available in cells to effectively inhibit UPF1 and avoid degradation of the histone mRNA during the S-phase. Very little SLBP is bound to UPF1 when the histone mRNA is actively translated, consistent with the observation that translation termination also plays an important role in recruiting UPF1 to the histone transcript[57]. In addition to UPF1, the SLBP N-terminal IDR also engages factors involved in nuclear export and translation of histone mRNA, and harbors sites for phosphorylation by the Cyclin A/CDK1 complex, which activates SLBP degradation[45,46,58]. These interactions occur at different stages of the life of histone mRNA and highlight the regulatory function of the N-terminal IDR throughout

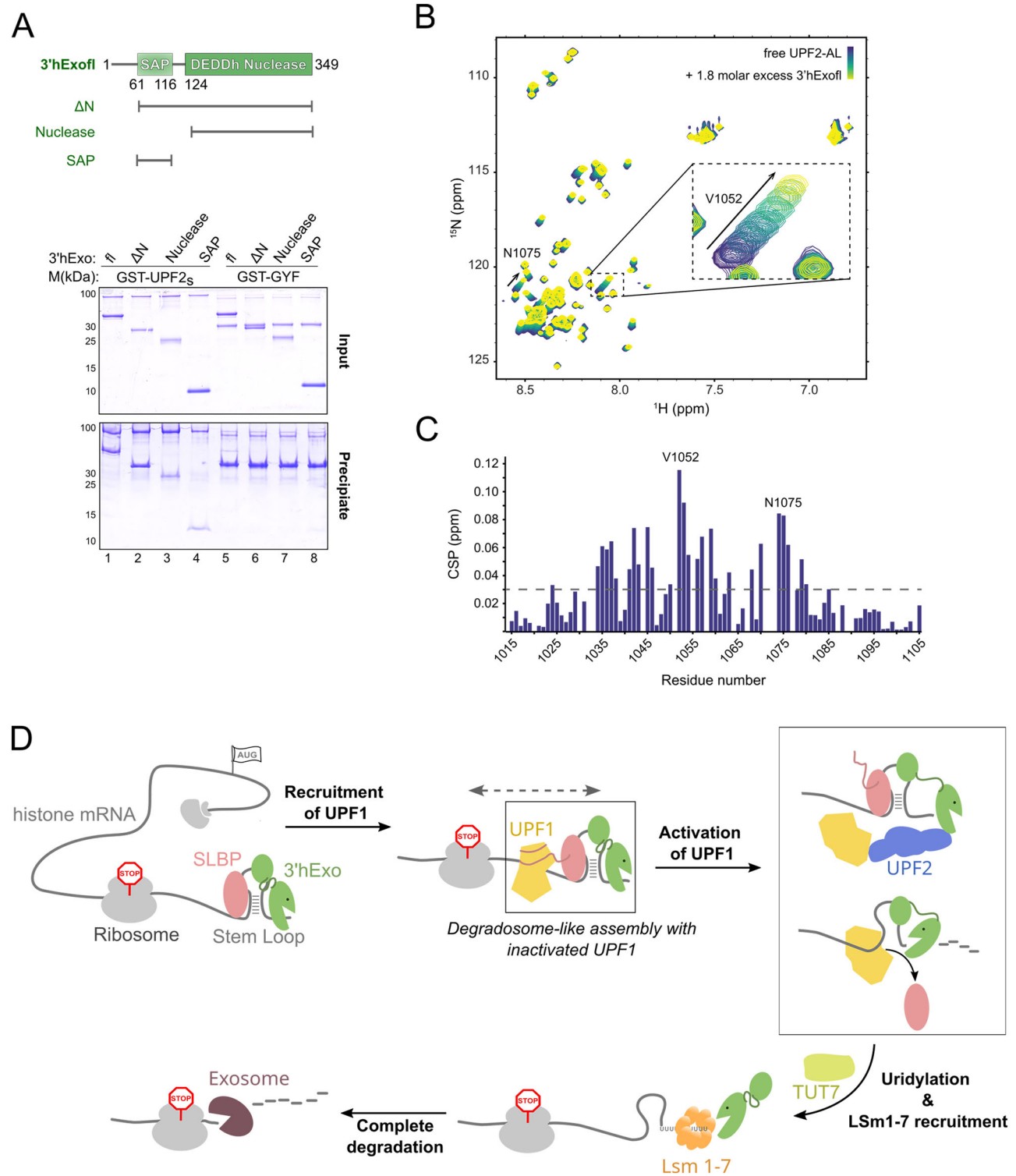

the entire process. The SLBP-UPF1 interaction positions UPF1 close to the histone SL, directly upstream of the SLBP-binding site. Since there is a requirement in vivo that the SL be close to the termination codon for histone mRNA decay[59], the recruitment of UPF1 in vivo might be functionally linked to recognition of a ribosome bound to the termination site (Fig. 7D). The initial degradation steps occur on polyribosomes, and there is a clear barrier to degradation 15 nts after the termination codon, which is likely the ribosome[59].

The UPF1-SLBP interaction is unusual in that it is mediated by the UPF1 helicase core and does not involve the CH domain or the intrinsically disordered C-terminal SQ region that bind other factors

involved in UPF1-mediated mRNA decay pathways, such as UPF2, STAU1, SMG5/7 and SMG6[26,47,50,60]. The only other known example of a protein–protein interaction mediated by the UPF1 helicase core is an intramolecular interaction with its own C-terminal SQ region[47]. Binding of the SQ region to the UPF1 helicase core inhibits its nucleic acid unwinding activity, possibly by restricting the conformational dynamics essential for a catalytic cycle. We speculate that binding of SLBP to the UPF1 helicase core might also have a similar effect. The binding and modulation of UPF1 by SLBP is reminiscent of the inhibition of UPF1 by the poly-pyrimidine tract binding protein PTBP1[32,61]. However, binding of PTBP1 to UPF1 triggers dissociation of UPF1 from

**Fig. 7 | Recruitment of UPF2 and the concerted regulation of UPF1 within the histone mRNP. A** UPF2 binds the SAP and nuclease domains of 3′hExo. Top: schematic representation of the domain organization of human 3′hExo. Residue numbers indicate domain boundaries. Protein variants used in this study are shown below. Bottom: GST-pulldown assay of GST-UPF2s (and GST-GYF as negative control) with 3′hExo variants. The negatively charged UPF2-AL likely recognizes positively charged pockets of the 3′hExo-SAP and nuclease domains. **B** ¹H-¹⁵N-HSQC NMR titration experiments of ¹⁵N-labeled UPF2-AL with increasing concentrations of 3′hExofl. The spectrum of free UPF2-AL is in blue while those recorded in presence of increasing concentrations of 3′hExofl are in progressively lighter shades of green. Two significant peak shifts are indicated by arrows. The inset shows a zoomed-in view of residue V1052 of UPF2-AL, exhibiting the largest CSP upon

addition of 3′hExo (see also Supplementary Fig. 7B, C). **C** Histogram of the chemical shift perturbations (CSP) of UPF2 AL upon titration of 3′hExofl plotted against the UPF2-AL sequence. The dashed line corresponds to the average CSP value obtained. **D** Model depicting the recruitment, function and regulation of UPF1 in the context of histone mRNA decay. UPF1 is recruited to the histone mRNA to form a degradosome-like assembly with SLBP and 3′hExo in the early stage of decay. The dashed arrow denotes the requirement for the SL to be within a certain distance of the termination codon, possibly for efficient recruitment of UPF1. Activation of UPF1 by UPF2 leads to displacement of SLBP, unwinding of the SL and degradation by 3′hExo. Initial decay by 3′hExo primes the SL RNA for oligouridylation by TUT7 and rapid mRNA decay, brought about by binding of the LSm1–7 complex and recruitment of bulk mRNA decay factors such as the exosome.

the target mRNA and protects it from decay[32], whereas binding of SLBP recruits and regulates UPF1 to promote histone mRNA decay. SLBP also acts as a roadblock for unwinding and degradation of the histone SL. We suggest that this role of SLBP is essential to ensure timely decay as UPF1 can spontaneously unwind the SL and trigger premature decay. Consequently, the main function of UPF1 in histone mRNA decay appears to be active displacement of SLBP, which in turn enables unwinding of the SL and eventually leads to its decay by 3′hExo.

Accumulation of distinct SL degradation intermediates in cells and in vitro studies suggests that 3′hExo is a distributive enzyme that dissociates from its substrate after each round of catalysis[10,16,34,39]. The SAP domain of 3′hExo recognizes the loop of the SL, rationalizing our observation that 3′hExo binds the intact SL and SL-decay intermediates with similar affinity[12]. We speculate that 3′hExo remain bound to the SL via its SAP domain, while UPF1 assists in repositioning of the 3′hExo nuclease domain on RNA by unwinding the SL to generate a small single-stranded overhang at the 3′ end. Unwinding activity of the histone SL by UPF1 is stimulated upon binding to UPF2, which is recruited to the histone mRNP through 3′hExo. Indeed, in every UPF1-mediated decay pathway investigated mechanistically thus far, UPF2 was shown to be anchored to the decay-RNP via a component other than UPF1[26,31,50]. Interestingly, UPF1 is a far more abundant protein in cells and is present in ~10-fold excess of UPF2[62], underscoring the importance of maintaining UPF2 near the helicase to ensure its rapid activation, particularly when timing of decay is crucial.

The composition of the initial mRNP in histone mRNA decay bears remarkable similarity to a degradosome, a multi-protein assembly that contains an RNA helicase, a ribonuclease and a sensor that dictates the timing of target mRNA decay[63]. We propose that the sensor in histone mRNA decay could be an event in the lifetime of the histone mRNA, such as translation termination or recruitment of the additional factors such as TUT7, the cytoplasmic LSm1-7 complex or the RNA exosome[36,64]. Alternatively, modification of the proteins involved, such as dephosphorylation of SLBP which weakens its binding to the histone SL or phosphorylation of UPF1 by SMG1, which was shown to play a role in histone mRNA decay, could be the sensor that triggers decay[13,16,65]. The molecular mechanism of how complete degradation is achieved, particularly coupling of decay to translation termination on the histone mRNA, and the signal generated when DNA replication is inhibited in S-phase or at the S-G2 phase transition of the cell cycle remain questions for further studies.

## Methods
### Cloning, expression and purification of recombinant proteins from *E. coli*
Plasmids expressing UPF1-CHh, UPF1-Hel, UPF1-CH and UPF2 variants (except UPF2-AL and UPF2sΔAL) were available from previous studies. The cDNA of human 3′hExo and its respective truncations as well as that corresponding to UPF2-AL were cloned into a modified pET28a vector with an N-terminal 6x His-tag, followed by HRV-3C and TEV protease cleavage sites, respectively, using ligation independent

cloning. The variant UPF2sΔAL was generated by round-the-horn-mutagenesis using the UPF2s plasmid as a template. The plasmid expressing UPF1-SLBP fusion was generated by two-step PCR, using the plasmids expressing UPF1-Hel and SLBPfl as templates. The UPF1 mutant, UPF1 AKS-HPA was also generated by two-step PCR using the UPF1-Hel plasmid and primers harboring the mutations. Primers used for cloning are listed in Supplementary Table 1. For expression of GST-tagged proteins, a vector containing an N-terminal tandem 6x His-GST tag followed by a TEV or 3C protease cleavage site was used. Proteins were expressed in *Escherichia coli* BL21 Star (DE3) pRARE (3′hExo) or BL21 Gold (DE3) pLysS (UPF1 and UPF2). Cells were grown in terrific broth (TB) under antibiotic selection at 37 °C for 18 h or until an $OD_{600nm}$ of ~2.5 was reached. The temperature was reduced to 18 °C and expression was induced by the addition of 0.1 mM IPTG (final concentration). Cells were harvested after 16–18 h and pelleted by centrifugation at $7460 \times g$ for 10 min. Cell pellets were resuspended in lysis buffer (50 mM Tris-HCl, pH 7.5, 500 mM NaCl, 10% (v/v) glycerol, 1 mM $MgCl_2$, 10 mM imidazole) and supplemented with DNase I and 1 mM PMSF. All buffers used for purification of UPF1-CHh were further supplemented with 1 μM $ZnCl_2$. Cells were lysed by sonication, and the lysate was clarified by centrifugation at $31,000 \times g$ for 45 min at 4 °C. The recombinant protein was enriched from the cell lysate using $Ni^{2+}$-affinity chromatography. Protein bound Ni-NTA beads were extensively washed with lysis buffer. In the case of UPF1-CHh, an additional wash step with chaperone wash buffer (50 mM Tris-HCl pH 7.5, 1 M NaCl, 10% (v/v) glycerol, 10 mM $MgCl_2$, 50 mM KCl, 2 mM ATP, and 10 mM imidazole) was included. Following a final wash step with low salt buffer (50 mM Tris-HCl pH 7.5, 150 mM NaCl, 10% (v/v) glycerol, 1 mM $MgCl_2$ and 10 mM imidazole). 6x-His tagged proteins were eluted from the Ni-NTA beads with elution buffer (50 mM Tris-HCl pH 7.5, 150 mM NaCl, 10% (v/v) glycerol, 1 mM (v/v) $MgCl_2$ and 300 mM imidazole). Proteins were further purified with a HiTrap Heparin Sepharose HP column (GE Healthcare) using heparin buffers A (20 mM Tris-HCl pH 7.5, 10% (v/v) glycerol, 1 mM $MgCl_2$, and 2 mM DTT) and B (20 mM Tris-HCl pH 7.5, 1 M NaCl, 10% (v/v) glycerol, 1 mM $MgCl_2$, and 2 mM DTT). Proteins were eluted from the column using a linear concentration gradient from 0 to 100% heparin buffer B over 20 column volumes. The final purification step comprised a SEC using Superdex 75 or Superdex 200 columns (Cytiva) in SEC buffer (3′hExo constructs: 20 mM HEPES buffer pH 7.5, 150 mM NaCl, 5% (v/v) glycerol, and 2 mM DTT; UPF1 and UPF2 constructs: 20 mM Tris-HCl pH 7.5, 150 mM NaCl, 5% (v/v) glycerol, 1 mM $MgCl_2$, and 2 mM DTT). Peak fractions were pooled, concentrated, flash frozen in liquid nitrogen and stored at −80 °C until further use. The purity of the proteins after each chromatography step was assessed by SDS-PAGE analysis and Coomassie staining.

For NMR spectroscopy proteins were recombinantly labeled with ¹⁵N and/or ¹³C stable isotopes in M9 minimal medium. *E. coli* cells were transformed with the desired expression plasmid, grown to an $OD_{600}$ nm of ~3.0 in TB media, spun down, and resuspended in an equal volume of minimal media. The cultures were then grown for 40 min at

37 °C to allow for recovery of the cells, followed by the addition of 0.5 g of $^{15}NH_4Cl$ and 8 g ($^{12}C$) or 2 g ($^{13}C$) of D-glucose in the case of $^{15}N$-labeled proteins or $^{13}C/^{15}N$ labeling, respectively. The cultures were induced with 0.5 mM IPTG at 18 °C for 16–20 h and centrifuged to collect the cell pellets, which were stored at −20 °C until further processing. Purification of isotopically labeled proteins was performed as described above with no modifications. The protein samples were dialyzed against 20 mM HEPES buffer (pH 7.0), 100 mM NaCl, 2 mM DTT, and concentrated to 100–500 μM using Amicon centrifugal filters of MWCO 3.5 kDa (Merck Millipore). The pH of all HEPES buffer stocks in every case was adjusted using sodium hydroxide. Protein concentrations were determined using the absorbance at 280 nm and the predicted molar absorptivity based on the protein sequence[55].

### Expression and purification of SLBP from insect cells

Bacmids for insect cell transformation and recombinant virus expressing full-length human SLBP and SLBP variants were generated using the Bac-to-Bac™ baculovirus expression system. Baculoviruses were generated by transfection of the bacmid in Sf9 cells (cultured in Sf-900 II media, Gibco) and further amplified by infection of Sf9 cells with a low-titer virus. For protein expression, High Five™ cells cultured in Express Five™ medium (Gibco) supplemented with glutamine were infected with 6xHis-TEV-SLBP viruses. Cells were harvested after 60 h by centrifugation at low speed. For purification, cells were resuspended in reduced salt buffer (10 mM Tris-HCl pH 7.5, 10 mM NaCl) supplemented with DNase I and 1 mM PMSF. After sonication, the cell lysate was supplemented with 5x Nickel binding buffer such that a final buffer composition of 20 mM Tris-HCl pH 7.5, 400 mM NaCl, 10 mM imidazole and 10% (v/v) glycerol was achieved. SLBP proteins were extracted from the lysate by $Ni^{2+}$-affinity chromatography and further purified by Heparin-affinity chromatography, as described above. A final SEC step was performed using a Superdex 75 column (Cytiva) in SEC buffer (20 mM Tris-HCl pH 7.5, 300 mM NaCl, 5% (v/v) glycerol and 2 mM DTT). Peak fractions were pooled, concentrated, flash frozen in liquid nitrogen and stored at −80 °C until further use. As with *E. coli* expressed proteins, an SDS-PAGE analysis was conducted after each step to monitor the purity of the proteins.

### CryoEM grid preparation, data acquisition and processing

For preparation of cryoEM grids, 17 μM of UPF1-CHh purified in 20 mM Tris-HCl and 100 mM NaCl was concentrated to 5.7 mg/ml and mixed with 1.22-fold molar excess of 12U-SL26 RNA and incubated for 20 min at 25 °C. The detergent n-octyl-β-D-glucoside was added to the sample to a final concentration of 0.15% (w/v) prior to grid preparation. ~4 μl of the mixture was applied to glow-discharged Quantifoil R1.2/1.3 holey carbon grids and plunged into liquid ethane using a Vitrobot Mark IV (Thermo Fisher) set at 10 °C and 100% humidity.

CryoEM data was acquired on an FEI Titan Krios G3i TEM operated at 300 kV equipped with a Falcon 3EC direct electron detector. Movies were taken for 40.57 s accumulating a total electron flux of 44 e-/Å2 in counting mode distributed over 33 fractions at a nominal magnification of 96,000x giving a calibrated pixel size of 0.819 Å/px. Automated data acquisition was conducted using EPU software (Thermo Fisher, Eindhoven, Netherlands). All image analysis steps were carried out using cryoSPARC[66]. Movie alignment was done with patch motion correction, CTF estimation was conducted with Patch CTF. Only micrographs with an information limit estimated during CTF determination below 9 Å, maximal in-frame motion below 20 px and total motion below 100 px were considered for further analysis. Class averages of manually selected particle images were used to generate an initial template for reference-based particle picking from 5074 micrographs. 3,102,505 particle images were extracted with a box size of 144 px and Fourier-cropped to 72 px for initial analysis. Reference-free 2D classification was used to select 666,034 particle images for further analysis. Ab initio reconstruction was conducted to generate an

initial 3D reference for heterogeneous 3D refinement. The final set of 300,222 particles was refined using non-uniform refinement after local motion correction and re-extraction of unbinned particles with a box size of 288 px, giving a final global resolution of 3.6 Å locally extending up to 2.8 Å.

### Model building, refinement and analysis

The crystal structures of human UPF1 (PDB-ID 2WJV) and of yeast Upf1 bound to RNA (PDB-ID 2XZL) were used to build the structural model reported here. Specifically, the individual UPF1 domains (CH, RecA1, RecA2, 1B, and 1C) from the human UPF1 structure and the $U_8$-RNA from the RNA-bound yeast Upf1 structure were placed in the cryoEM reconstruction and adjusted by rigid body refinement using Coot (version 1.1.11)[67]. Missing loop regions of UPF1 were manually connected. The crystal structure of histone SL (PDB-ID 4L8R) guided initial placement of the SL region in the electron density. The single stranded 5′ end was further extended according to the cryoEM reconstruction. The model was refined by iterative rounds of real space refinement as implemented in PHENIX (version 1.21.2_5419)[68,69] and manual adjustment in Coot. The structural model was evaluated with MolProbity (version 4.5.2)[70]. Structure figures were prepared with PyMOL (version 2.1, Schrödinger) and ChimeraX (version 1.9)[71].

### GST pulldown assays

8 μg GST-tagged bait protein was incubated with 8 μg prey protein for 1 h, at 25 °C in GST-pulldown buffer (20 mM HEPES buffer pH 7.5, 70 mM NaCl, 0.1% NP-40 and 10% (v/v) glycerol). Samples were washed three times with GST-pulldown buffer before elution with GSH elution buffer (30 mM Tris pH 7.5, 150 mM NaCl, 1 mM $MgCl_2$, 20 mM glutathione, 0.1% NP-40, 14 % (v/v) glycerol, 2 mM DTT). Input and elution samples were resolved on an SDS PAGE gel and stained with Coomassie Brilliant Blue. GST-GYF was used as a negative control to mimic hydrophobic interaction interfaces that could non-specifically capture aggregated SLBP proteins.

### Size exclusion chromatography (SEC)

For complex formation between UPF1 and SLBP, 1000 pmol of each protein and synthetic histone SL26 RNA (Integrated DNA Technologies, IDT), added in 1.2-fold molar excess wherever mentioned, were mixed to a final volume of 50 μl in analytical SEC buffer (20 mM HEPES buffer pH 7.5, 75 mM NaCl, 5% (v/v) glycerol, 1 mM $MgCl_2$, 2 mM DTT). For analysis of 3′hExo and UPF2 interactions, 1000 pmol of 3′hExo and 2000 pmol of UPF2-AL were used for the analysis. Samples were incubated overnight at 4 °C and resolved on a Superdex 200 Increase 3.2/300 column (GE Healthcare). The peak fractions were analyzed by SDS-PAGE, followed by staining with Coomassie brilliant blue. Protein-RNA complexes were additionally analysed on a 10% urea-PAGE gel and stained with ethidium bromide.

The 3′hExo-UPF1-UPF2 complex was reconstituted on a preparative scale, using UPF1-CHh, UPF2s and a variant of 3′hExo lacking the first 60 residues (3′hExoΔN). The proteins were mixed in equimolar amounts, incubated overnight at 4 °C and resolved on a Superdex 200 16/600 column using SEC buffer (20 mM HEPES buffer pH 7.5, 100 mM NaCl, 2% glycerol, 1 mM $MgCl_2$, 2 mM DTT). The peak fractions were analysed by SDS-PAGE as above.

### Crosslinking mass spectrometry

To reconstitute the UPF1-SLBP-SLRNA complex on a preparative scale for CXMS, UPF1-Hel, SLBP-N, and SL RNA were mixed at a molar ratio of 1.2:1:1.2, with the total protein amounting to ~8 mg. The mixture was diluted to 1–2 mg/mL and incubated at room temperature for 30 min, followed by dialysis against CXMS-SEC buffer (20 mM Tris-HCl pH 7.5, 75 mM NaCl, 5% glycerol, 1 mM $MgCl_2$, and 2 mM DTT) overnight at 4 °C. The mixture was concentrated and injected onto a Superdex 200 10/300 Increase column to separate the ternary complex from

individual components and sub-complexes. The peak fractions were analysed by SDS-PAGE and Coomassie staining to detect proteins as well as urea-PAGE and ethidium bromide staining to detect RNA. Fractions corresponding to the ternary protein-RNA complex were pooled and concentrated to 1.5 mg/mL for crosslinking.

For crosslinking, the complex was first exchanged into CX-buffer (20 mM HEPES, pH 7.5, 75 mM NaCl, 5% (v/v) glycerol, 1 mM $MgCl_2$). 10 µg aliquots of the protein complex were initially crosslinked with 0.2, 0.5, 1, and 2 mM BS3 (Thermo Scientific) to optimize the minimum concentration of crosslinker necessary to achieve efficient cross-linking. A control reaction without a crosslinker was performed in each case. The final crosslinking reaction was carried out with 1 mM BS3 for 30 min at room temperature and quenched with 50 mM Tris. Samples were separated by SDS-PAGE (NuPAGE 4–12% gradient gel, Invitrogen). The crosslinked complex was cut out of the gel and separated into three pieces. Excised gel pieces were then subjected to in-gel tryptic digest. Samples were reduced with 10 mM DTT and alkylated with 55 mM iodacetamide and subsequently digested with trypsin (sequencing grade, Promega) at 37 °C for 18 h. Extracted peptides were dried in a SpeedVac Concentrator and dissolved in loading buffer composed of 4% acetonitrile and 0.05% TFA. Samples were subjected to liquid chromatography mass spectrometry (LC-MS) on a QExactive HF-X (Thermo Scientific). Peptides were loaded onto a Dionex UltiMate 3000 UHPLC+ focused system (Thermo Scientific) equipped with an analytical column (75 µm × 300 mm, ReproSil-Pur 120 C18-AQ, 1.9 µm, Dr. Maisch GmbH, packed in house). Separation by reverse-phase chromatography was done on a 60-min multi-step gradient with a flow rate of 0.3–0.4 µl min⁻¹. MS1 spectra were recorded in profile mode with a resolution of 120 k, maximal injection time was set to 50 ms and AGC target to 1e⁶ to acquire a full MS scan between 380 and 1580 m/z. The top 30 abundant precursor ions (charge state 3–8) were triggered for HCD fragmentation (30% NCE). MS2 spectra were recorded in profile mode with a resolution of 30 k; maximal injection time was set to 128 ms, AGC target to 2e⁵, isolation window to 1.4 m/z, and dynamic exclusion was set to 30 s. Raw files were analyzed via pLink2.3.5 to identify crosslinked peptides, with standard settings changed as follows—peptide mass: 600–10,000, precursor tolerance: 10 ppm, fixed modification: Carbamidomethyl [C], and variable modification: Oxidation [M][72]. False discovery rate was set to 1% and results were filtered by excluding crosslinks supported by only one crosslinked peptide spectrum match (Supplementary Table 2). The interaction network for the ternary complex was illustrated via xiNET[73]. The crosslinking was performed three times on the same complex reconstituted by SEC (technical triplicates).

### CRISPR/Cas9 mediated SLBP exon 2 deletion and RT-PCR analysis

SLBP exon 2 was deleted in HCT116 cells using CRISPR/Cas9 as described earlier[35]. Four sgRNAs targeting the first and second introns flanking exon 2 were designed using CRISPOR. These sgRNAs were then synthesized with "ccgg" and "aaac" modifications at the 5′ ends of the forward and reverse strands, respectively, to match the sticky ends of the vector after annealing. The sgRNAs were phosphorylated and ligated into a BsaI-digested sgRNA vector (Addgene #51133). HCT116 cells were cultured in McCoy's 5 A medium (Corning, 10-050-CV), and approximately $0.4 \times 10^6$ cells were seeded in six-well plates. Transfection was performed 24 h later. The correctly sequenced sgRNA plasmids were co-transfected with Cas9 (Addgene #44758) and GFP (pmaxGFP from Lonza) plasmids into cells using jetOPTIMUS (Genesee Scientific). GFP positive single cells were isolated with the CellRaft Air® System and cultured in 96-well plates to obtain clonal cell lines. Genomic PCR, with primers designed to amplify a 500 bp region around the target sites, was used to confirm the presence of the mutation. Positive PCR products were sequenced (Eton Bioscience) and cloned into a pJET vector (Thermo Fisher, K1231) for allelic sequence determination.

To confirm exon 2 deletion at the mRNA level, RT-PCR was performed. RNA was extracted from cells, and cDNA was synthesized using a reverse transcriptase kit (Applied Biostems 4387406). PCR amplification was carried out with a forward primer located in exon 1 and a reverse primer spanning the exon 4-exon 5 junction. This strategy ensured that only the mRNA-transcribed cDNA was amplified, while any genomic DNA contamination was eliminated by the primer design. The PCR products were analyzed by agarose gel electrophoresis to verify the absence of exon 2 in SLBPΔ2 cells.

### Western blot

To verify exon 2 deletion at the protein level, Western blotting was performed as described by Holmquist and co-workers[35]. Cells were collected and lysed using a buffer containing 0.5% NP-40, 50 mM Tris-HCl, pH 7.5, 300 mM NaCl, and protease inhibitors (Roche, 11836170001). The lysates were incubated on ice for 20 min, followed by centrifugation at 4 °C. Protein concentration was determined using a BCA protein assay. A total of 10 µg of protein was separated on a 4–12% gradient gel and transferred to a nitrocellulose membrane (Thermo REF 88018) at 100 V for 1 h. The membrane was blocked in 5% non-fat milk in phosphate-buffered saline supplemented with 0.1% Tween-20 for 1 h and incubated overnight at 4 °C with primary antibody in PBST. After washing, the membrane was incubated with a secondary antibody for 1 h at room temperature. The membrane was washed again, and protein signals were developed using ECL for 3 min. Protein expression was visualized using a KwikQuant imaging system (Kindle Bioscience).

### Northern blot

For study of histone mRNA decay, wt and SLBPΔE2 cells were treated with 5 mM HU (hydroxyurea, Sigma) for 20 and 40 min. Total RNA was extracted from untreated and HU-treated cells using Trizol reagent (Ambion, 15596026). RNA was quantified using a Nanodrop spectrophotometer. For Northern blotting, 5 µg of RNA was denatured by incubation at 95 °C for 5 min and then separated by electrophoresis on a 6% AccuGel. RNA was transferred to a membrane (Cytiva RPN303B) using cold 0.5x TBE at 30 V for 40 min. The membrane was dried for 1 h and cross-linked with UV light at 1200 µJOULES × 100 using a Strata-linker 2400 system. For hybridization, the membrane was pre-hybridized in hybridization buffer (Invitrogen, AM8670) at 42 °C for 30 min. Probes were synthesized from purified PCR templates using random priming, denatured at 95 °C, and hybridized to the membrane overnight at 42 °C. The membrane was washed twice with 2% saline sodium citrate (SSC) and once with 0.01% SSC before being scanned for signals using a Typhoon imaging system (GE Healthcare).

### In vitro transcription of RNA substrates

For precise RNA 3′ ends, histone SL RNAs were in vitro transcribed from a DNA template where the HDV-ribozyme sequence was fused to the 3′ end of the template for SL RNA (Supplementary Fig. 1B and Supplementary Table 1). DNA template sequences were cloned into a pJet1.2/blunt cloning vector using the CloneJET PCR-cloning kit (ThermoFisher). PCR-amplified double-stranded (ds) DNA template was incubated with T7 RNA polymerase (purified in-house) together with 16 mM NTPs, 50 mM DTT, 20 mM $MgCl_2$, 2 mM Spermidine, 6.4% PEG 8000, and pyrophosphatase enzyme (purified in-house) in a 200 mM Tris-HCl pH 8 buffer for 4-6 h at 37 °C. The reaction was quenched with EDTA prior to gel purification of the transcribed RNA. The reaction mix consisting of different RNA fragments was resolved on a 10% denaturing PAGE, from which the gel fragment corresponding to the auto-cleaved SL RNA was excised. Gel pieces were crushed and soaked overnight in RNA elution buffer (20 mM Tris-HCl, 300 mM sodium acetate, 2 mM EDTA). Recovered RNA in solution was buffer-exchanged into nuclease-free water (HyPure water, Cytiva) and concentrated using a 3-kDa molecular weight cut-off concentrator

(Merck). To remove residual buffer components, RNA was precipitated out of solution using ethanol and redissolved in nuclease-free HyPure water. The purified RNA was stored at −20 °C until further use.

## Unwinding assay and analysis

Unwinding assays were based on those described by Fritz and co-workers[32] and were adapted for our purposes. RNA substrates were in vitro transcribed from PCR-amplified dsDNA templates, following the protocol above. Substrates for unwinding assays were not transcribed as precursors fused to HDV-ribozyme. 5′-Alexa Fluor488-labeled DNA that was used to create the RNA:DNA duplex and Black-hole Quencher 1 (BHQ1)-labeled DNA (used as a trap) was purchased from IDT. Sequences of all nucleic acids used in this experiment (including the DNA template for in vitro transcription) are provided in Supplementary Table 1. For duplex annealing, the respective RNA and fluorescent DNA were mixed in a 11:7 ratio to a final concentration of 864 nM RNA and 543 nM DNA together with 2 mM magnesium acetate in 1x unwinding buffer (50 mM MES buffer pH 6.5, 50 mM potassium acetate, 0.1% NP-40). The RNA:DNA duplex was incubated at 96 °C for 4 min, snap-cooled on ice for 30 min and used the same day. For each replicate, a total reaction volume of 40 μl was prepared with 1x unwinding buffer supplemented with 2 mM magnesium acetate and 2 mM DTT. 7.5 pmol of DNA:RNA duplex were mixed with 12 pmol of UPF1-Hel or UPF1CH-h protein, and wherever mentioned with12 or 20 pmol of the respective SLBP protein and/or 24 pmol UPF2$_S$ in a volume of 22 μl. The reaction was incubated for 30 min at 25 °C. 35 pmol of BHQ1 DNA quencher was added, following which the reaction was transferred to a black, flat-bottomed, 384-well plate (Corning). Negative controls contained 1x unwinding buffer instead of protein or RNA. To start the reaction, 16 μl of 2 mM ATP in 1x unwinding buffer was added to each sample using the injector module of a Spark multimode microplate reader (Tecan). Fluorescence was monitored over a period of 60 min in 10 s intervals. Two technical duplicates were performed for each condition. The values were corrected for baseline fluorescence by subtracting the initial fluorescence reading immediately following the addition of ATP. This was then normalized to the baseline-corrected maximum fluorescence value of UPF1-Hel for each data set to obtain values for fraction unwound, which are plotted as a function of time. The data were fitted to a two-step decay model using Graph-Pad Prism (version 10.2.3) to obtain first-order rate constants for the fast and slow phases of unwinding ($k_{fast}$ and $k_{slow}$, Fig. 1B, Supplementary Fig. 5A–C). Statistical significance was determined by unpaired t-tests.

## Fluorescence anisotropy assay

To determine the binding affinities of 3′hExo and SLBP to SL RNA and SL degradation intermediates by fluorescence anisotropy assay, synthetic RNAs labeled with 6-FAM at their 5′ end were procured from IDT (Supplementary Table 1 for RNA sequences). The respective RNA was incubated with increasing amounts of SLBPfl or 3′hExofl. A 45 μl binding-reaction containing RNA and protein was prepared in 1x fluorescence anisotropy buffer (20 mM HEPES pH 7.5, 100 mM NaCl, 0.5 mM EDTA, supplemented with bovine serum albumin to a final concentration of 100 μg/ml). The reactions were incubated for 30 min at room temperature. 40 μL of each sample was transferred to a black, flat-bottomed, 384-well plate (Corning). The negative controls were prepared with 1x fluorescence anisotropy buffer in place of the protein. Fluorescence polarization was measured with a multi-modal Tecan plate reader (Spark) at 25 °C. Technical duplicates were prepared and measured for each protein concentration. Fluorescence anisotropy values were corrected for background by subtracting the value obtained for the negative control lacking protein and subsequently normalized to the maximum anisotropy value. Each experiment was performed three times; values from the three independent

experiments were averaged and fitted to an equation representing one site-specific binding with a Hill slope using Prism software (GraphPad).

## Degradation assay

To monitor the degradation of SL RNA by 3′hExo, a time course experiment was performed using RNA labeled at its 5′ end with $^{32}$P, where the degradation products of the RNA were resolved on a denaturing PAGE. 0.1 nM radiolabelled RNA was incubated with 200 nM 3′hExofl (or 400 nM 3′hExofl for 2 × 3′hExo), 200 nM cold RNA and where indicated, with 200 nM UPF1-Hel, in 1x degradation buffer (20 mM HEPES buffer pH 7.5, 50 mM NaCl, 5 % (v/v) glycerol, 2 mM DTT) on ice at 4 °C. The degradation reaction was initiated by the addition of 5 mM MgCl$_2$ and 1.7 mM ATP (final concentrations) and conducted at 25 °C. 10 μl were removed from the reaction mixture at the indicated time points and were quenched with an equal volume of 2x RNA dye. For the 0 min time point, 10 μl was removed prior to the addition of MgCl$_2$ and ATP. Degradation products were resolved on a 15% denaturing PAGE. The RNA was visualized via phosphorimaging as described above for EMSAs.

## NMR spectroscopy

NMR experiments were collected on a Bruker Avance 600, 700, and 800 MHz spectrometers at 298 K. The NMR samples contained 10% D$_2$O as a lock agent and 0.2% sodium azide for increased sample stability. Titration experiments were performed using isotopically $^{15}$N-labeled UPF2-AL or UPF2-Alm (residues 1024–1085), the latter used for $K_D$ determination (see below). The backbone assignments of UPF2-AL ($^{13}$C$^{\alpha}$, $^{13}$C$^{\beta}$, $^{15}$N, and $^1$H$^N$) were obtained with $^{15}$N/$^{13}$C-labeled samples and standard $^1$H-$^{13}$C-$^{15}$N scalar correlation experiments with apodization weighted sampling[74,75]. The secondary structure propensity obtained from NMR chemical shifts was determined using the *M*otif *I*dentification from *C*hemical *S*hifts (MICS) algorithm[76].

To monitor changes occurring in UPF2 upon addition of 3′hExo, $^{15}$N-HSQC spectra were recorded after stepwise addition of unlabeled proteins, either full-length or containing the SAP (63–124) or nuclease (124–349) domains of 3′hExo. The resulting spectra were analyzed using NMRPipe and NMRFAM-Sparky[77,78]. The CSP values in the presence of varying amounts of 3′hExo were calculated as $\Delta\delta = [(0.14\Delta\delta_N)^2 + (\Delta\delta_H)^2]^{1/2}$, where $\Delta\delta_N$ and $\Delta\delta_H$ are the changes in chemical shift for $^{15}$N and $^1$H$^N$, respectively. To determine the $K_D$, twelve well-resolved amide peaks exhibiting fast-exchange behavior upon addition of 3.75 molar excess of 3′hExofl to UPF2-ALm were tracked. The CSP values were plotted as a function of added 3′hExofl, and the resulting titration curves fit with GraphPad Prism to the equation for a one-to-one binding isotherm as described by Perrez-Borrajero and co-workers[79]. The resulting mean value and standard deviation are reported.

## Reporting summary

Further information on research design is available in the Nature Portfolio Reporting Summary linked to this article.

# Data availability

The data supporting the findings of this study are available from the corresponding authors upon request. The three-dimensional cryoEM density maps of the UPF1 bound to SL26 have been deposited in the EM database with the accession code EMD-53417. Corresponding structural coordinates are available in the Protein Data Bank under the accession code 9QWN. The chemical shifts of UPF2 have been deposited in the Biological Magnetic Resonance Data Bank (BMRB) under accession code 53039 (10.13018/BMR53039). The mass spectrometry proteomics data have been deposited to the ProteomeXchange Consortium via the PRIDE partner repository with the dataset identifier PXD070077. Source data for the figures and

Supplementary Figs. are provided as a Source Data file. Source data are provided with this paper.

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

## Acknowledgements

We thank Rhese D. Thompson and Qi Zhang for sharing their expertise on in vitro transcription, Lea S. Pommerening for help with establishing nucleic acid unwinding assays, Markus C. Wahl for access to the insect cell culture facility, Florian Heyd for critically reading the manuscript and members of the Chakrabarti lab for helpful discussions. We acknowledge access to electron microscopy equipment at the core

facility BioSupraMol of Freie Universitat Berlin, supported through grants from the Deutsche Forschungsgemeinschaft and the State of Berlin for large equipment according to Art. 91b GG (INST 130/1014-1 FUGG, INST 335/588-1 FUGG, INST 335/589-1 FUGG, and INST 335/590-1 FUGG). Access to the Tecan Spark plate reader and Biacore™ X100 was also obtained through BioSupraMol. We are grateful for access to high-performance computing resources at the Zuse Institut Berlin. S.C., J.H., and H.U. are supported by grants from the Deutsche For-schungsgemeinschaft (DFG) and acknowledge funding through the priority program SPP1935 (CH1245/3-1 and 3-2 and CH1245/6-1 to S.C.). S.C. is additionally supported by the Heisenberg program of the DFG (CH1245/5-1). H.U. was also supported by DFG-funded consolidated research consortium SFB860. C.P.-B. was supported by the EMBL Interdisciplinary Postdoc (EI$_3$POD) Program fellowship under Marie Sklodowska-Curie Actions COFUND (grant no. 664726). W.F.M. is sup-ported by grants from the NIH.

## Author contributions

A.M.A, G.X., T.D., S.L., and S.C.: protein purification, biochemical and biophysical experiments; T.H.: acquisition and processing of cryoEM datasets; A.M.A, B.L., and S.C.: Model building and interpretation of structure; W.H. and W.F.M.: analysis of histone mRNA decay in wild-type and CRISPR/Cas9 edited cells; C.P.-B. and J.H.: NMR spectroscopy and data analysis. J.B. and U.H.: crosslinking mass-spectrometry and data analysis; N.V., V.N., and C.K.: additional biochemical and biophysical experiments. A.M.A., W.F.M., and S.C.: experimental design and manu-script preparation; all authors: manuscript editing.

## Funding

## Competing interests

The authors declare no competing interests.

## Additional information

[1]Institute of Chemistry and Biochemistry, Department of Biology, Chemistry and Pharmacy, Freie Universität Berlin, Berlin, Germany. [2]Integrated Program for Biological and Genome Sciences, University of North Carolina, Chapel Hill, NC, USA. [3]Department of Biochemistry and Biophysics, University of North Carolina, Chapel Hill, NC, USA. [4]Molecular Systems Biology Unit, European Molecular Biology Laboratory (EMBL), Heidelberg, Germany. [5]Research Group Bioanalytical Mass Spectrometry, Max Planck Institute for Multidisciplinary Sciences, Göttingen, Germany. [6]Bioanalytics, Department of Clinical Chemistry, University Medical Center Göttingen, Göttingen, Germany. [7]Institute of Chemistry and Biochemistry, Laboratory of Structural Biochemistry, Freie Universität Berlin, Berlin, Germany. [8]Research Center of Electron Microscopy and Core Facility BioSupraMol, Freie Universität Berlin, Berlin, Germany. [9]Chair of Bio-chemistry IV, Biophysical Chemistry, University of Bayreuth, Bayreuth, Germany. [10]Present address: Gene Center and Department of Biochemistry, Ludwig-Maximilians-Universität München, Munich, Germany. [11]Present address: Institute of Immunology, Christian-Albrechts-Universität zu Kiel & Uni-versitätsklinikum Schleswig-Holstein (UKSH), Kiel, Germany. [12]Present address: Vienna BioCenter Core Facilities, Vienna, Austria. [13]Present address: Nuclear Dynamics and Cancer Program, Cancer Epigenetics Institute, Fox Chase Cancer Center, Philadelphia, PA, USA. [14]These authors contributed equally: Alexandrina Machado de Amorim, Guangpu Xue. ✉e-mail: chakraba@zedat.fu-berlin.de

