## [Peer Review File · Nature Communications]

Mechanistic insights into recruitment and regulation of the RNA helicase UPF1 in replication-dependent histone mRNA decay

Corresponding Author: Professor Sutapa Chakrabarti

Version 0:

Reviewer comments:

Reviewer #1

(Remarks to the Author)

In this manuscript the authors Machado de Amorim and Xue et al. use structural biology and biochemical approaches to investigate the molecular mechanism behind the first steps in replication-dependent histone mRNA decay and their regulation by the RNA helicase UPF1 in metazoans. This is an intriguing problem in mRNA biology since replication dependent histone mRNAs belong to the few eukaryotic mRNAs that are not polyadenylated at their 3' end but rather carry a 3' stem loop (SL) instead (other non-polyadenylated mRNAs include yeast mitochondrial mRNAs). This 3' stem loop motive has been known for several decades and controls both mRNA stability but also decay. Machado de Amorim and Xue et al. in well designed experiments carefully dissect the sequence of individual interactions of the 3'SL with proteins that partake in the RNA degradation process, including UPF1, SLBP, UPF2 and 3'hExo. This allows the authors to propose a mechanistic model where the ternary 3' stem loop:SLBP:3'hExo complex recruits both UPF1 and UPF2 via direct protein-protein interactions. UPF2-mediated activation of the UPF1 helicase activity leads to destabilisation of the stem loop:SLBP interaction, which in turn resolves the road block that RNA-bound SLBP imposes on 3'hExo activity. The 3'hExo RNase is then able to degrade the SL which subsequently initiates downstream histone mRNA degradation steps. The work for the first time offers intriguing insights into the mechanism with which UPF1 governs a specific mRNA degradation pathway and into how UPF1's own RNA helicase activity is tightly controlled by the cross play of the regulatory factors SLBP and UPF2 in the histone mRNP.

While the results reported are of interest to a broad readership and the data are both beautiful and carefully interpreted some of the datas' presentation and documentation, especially concerning the cryoEM analysis, leaves much to be desired. The issues raised below should therefore be carefully addressed before the manuscript is ready for publication in Nature Communications.

Major points:

-The authors should document their cryoEM processing pipeline and the quality of their final 3D reconstruction more carefully. A detailed processing scheme outlining the individual steps should be included in the supplementary figures of the manuscript. This is especially interesting since other conformational states of the UPF1 and UPF1-SL complex might have been resolved in the data set. Additionally it is currently not clear from the included structural figures that the resolution of the cryoEM map really reaches below 3 Å as reported. The authors should therefore include a gallery of well resolved map areas with the respective model parts (below 3 Å one expects to resolve medium size amino acid side chains and RNA bases individually). If this is not possible the authors should caveat their reported resolution estimates more carefully. An map-vs-model FSC curve should be included as a supplementary figure panel and reported in table1 too. All of this is customary in reporting single particle cryoEM reconstructions.

-The authors have carefully mapped the protein-protein interactions between UPF2 and hExo3' and between UPF1 and SLBP biochemically with several approaches. Have they attempted to predicted the structures of these minimal interacting regions and/or full lengths proteins using AlphaFold or similar machine learning-based structure prediction tools? This could potentially offer additional critical insights into the structural organisation of the histone mRNPs and could help to integrate

previously published data and here reported insights into an overall structural picture further strengthening this study.

Minor points:

-FigS1 A: if this reviewer understands the experimental set up correctly the authors inhibited the activity of hExo3' by chelating divalent cations via EDTA in the reaction buffer to measure the association between SL intermediates and hExo3'. Is this correct or how did the authors inhibit RNase activity/control for robust measurements in these assays (especially with regards to the SL19 substrate)? One would expect a clarifying statement to this end in either the main text or the figure legend.

-FigS1 D: scale bar is missing in cryoEM micrograph; please use 100 Å to make the display comparable to the scale bar in the 2D class averages

-FigS1 G: it is highly doubtful that the quoted resolution estimates carry any meaning beyond the 1st decimal. Please correct the display and the figure legend accordingly.

- table 1: please also quote the per-chain correlation coefficient between the map and the model. This is relevant since the model contains both RNA and protein moieties.

- The authors should explain how they filtered the cryoEM micrographs from 5,498 recorded to 5,074 used in subsequent processing. What criteria were used?

- It is currently unclear how many technical and/or biological replicates were used in the cross linking ms experiments reported in Figure 3. The authors should explicitly clarify this in the figure legend of Figure 3 and in the respective material and methods section.

- Similar to the layout of the informative protein domain organisation schemes it would be helpful if the authors provided such schemes for the different SL intermediates/constructs

page 4; line 20 : "... functional mRNA decay.." it is unclear to the reader what the authors mean by "functional" here. Is there "non-functional" decay?

page 8 line 7/8 citation/pdb code missing for SL:SLBP structure

page 9 line 9/10 why do the authors perceive there to be a bigger stimulatory effect of active unwinding on hExo3' activity in cells compared to their assay set up. Isn't a greater effect through direct UPF1 recruitment via specific protein-protein interaction equally if not more plausible?

page 14 line 17 there is an "in" missing; it should read "... and in previous studies"

page 15 line 16/17 does this sentence refer to a figure or a previous paper? please clarify and add respective reference if applicable.

Material and methods "Model building..." part (starting page 24): the authors say they used both a previously published yeast as well as a human UPF1 structural model to build the UPF1 model reported here. However they do not specify for what or why the yeast model was used or how they proceeded with this. This should be clarified.

Reviewer #2

(Remarks to the Author)

Machado de Amorim et al. describe the molecular mechanisms initiating histone mRNA decay by showing that UPF1, SLBP, and 3'hExo form a degradosome-like complex. Using cryoEM, the authors reveal that UPF1 binding alone distorts and partially melts the histone stem-loop (SL) RNA, priming it for degradation by 3'hExo even in the absence of ATP. SLBP directly engages the UPF1 helicase core via its N-terminal intrinsically disordered region, attenuating UPF1 activity and preventing premature unwinding. The SLBP-UPF1 interaction is essential for efficient decay in cells, as deletion of the UPF1-binding region in SLBP delays mRNA degradation. Furthermore, the study identifies 3'hExo as the recruitment platform for the UPF2 activator, which enhances UPF1 helicase activity via its acidic linker. Together, these findings establish a mechanistic framework for how protein-protein and protein-RNA interactions coordinate regulated degradation of replication-dependent histone mRNAs.

This is an overall very interesting and timely study. The results are very clear and well-controlled, and I only have few comments.

Major comments:

1) experiments in the SLBP Δ E2 cell line: this is a very interesting tool. if Antibodies for IP are available, it would be great if the authors could test by IP (WB or MS) whether the interaction between SLBP and UPF1 is lost (if possible, in the absence and presence of inhibitor of DNA replication)

2) Figure 5A/B: it would be great if the authors could add the N-terminal IDR (without RBD) as a control

Minor comments:

- 3) For Figure 2A / Supplementary Figure 2A: it would be helpful to see the two structures side-by-side
- 4) Figure 2C: please add a quantification
- 5) Figure 3C: can the authors comment on the stoichiometry of UPF versus SLBP in their stable complex? From the gel, I am not sure it is 1:1
- 6) figure 5 and 6: does the fraction unwound depend on the concentration of SLBP protein? What is the molar ratio of SLBP versus UPF1 in cells?

Reviewer #3

(Remarks to the Author)

This manuscript presents a detailed mechanistic study of replication-dependent histone mRNA decay, focusing on the molecular interactions among UPF1, SLBP, 3'Exo, and UPF2. The authors employ structural, biochemical, and reconstitution assays to dissect the role of UPF1 in stem-loop remodeling and decay initiation. The work provides valuable insights into the architecture and regulation of a degradosome-like complex that controls histone mRNA turnover. The structural data and in vitro assays are thoughtfully designed and interpreted. However, while the molecular reconstitution is convincing, there are important limitations regarding the physiological relevance of the findings, particularly the lack of in vivo or cellular validation of the proposed dynamic regulatory mechanism. Below are detailed comments.

Below are the list of key concerns;

1. The study aims to elucidate how UPF1, SLBP, 3'Exo, and UPF2 coordinate to drive histone mRNA decay. While the structural and biochemical data convincingly show how these components interact and modulate each other's activity in vitro, the cellular context in which these interactions occur is not addressed. Given that histone mRNA decay is a rapid, tightly regulated, and cell cycle-linked process, the lack of time-resolved or phase-specific cellular experiments is a key limitation.

In particular, the authors propose that the SLBP-UPF1 interaction both recruits and regulates UPF1 activity to prevent premature histone mRNA decay. While this model is supported by in vitro structural and biochemical data, the study does not include time-resolved cellular experiments that examine how these interactions change in a physiologically dynamic context, such as during S-phase progression or in response to replication stress.

A major missing element is the role of UPF1 phosphorylation, which is a well-established regulatory mechanism controlling its helicase activity and interaction with decay cofactors. In vivo, UPF1 is phosphorylated in a cell cycle-dependent manner, which influences its recruitment, catalytic state, and decay initiation potential. However, all in vitro experiments in this study were performed with recombinant, unphosphorylated UPF1, thereby excluding this essential regulatory layer.

2. Figures 1B and 5A-C present unwinding kinetics that are central to the paper's mechanistic model — namely, that UPF1-mediated stem-loop unwinding is modulated by SLBP and its domains. However, the interpretation of these data lacks statistical rigor, and the trends reported are not sufficiently substantiated with quantitative analysis.

In Figure 1B, the unwinding efficiency of SL RNA appears to be greater than that of the linear RNA. Yet it is unclear whether this difference is statistically significant or simply due to experimental variation.

In Figures 5A-C, SLBP inhibits UPF1 unwinding via its N-terminal IDR, while SLBP-RBD shows no effect on linear RNA (5B) but unexpectedly slows early unwinding and enhances it later on SL RNA (5C), a biphasic effect that requires clearer explanation and statistical validation.

Minor comments:

1. Figure 1D/E: The rationale for introducing a polyuridine (polyU) sequence upstream of the SL in the RNA construct should be clarified.
2. Figure 3A: The role of GST-GYF as a negative control in the pulldown assay should be explained.
3. Figure 3C: The interpretation of peak 4 as aggregated SLBP-N needs clarification. Since both SLBP-N and UPF1 show smearing, it is recommended to distinguish specific aggregation from nonspecific background.
4. Figure 3D: Clearly labeling the domain architecture of both UPF1 and SLBP in the figure would improve interpretability of the crosslinking results.
5. Figure 4C: The authors conclude that deletion of SLBP residues 29-51 impairs UPF1 recruitment. However, this region might also mediate interactions with other unknown cellular cofactors. The possibility of indirect effects should be acknowledged.

Version 1:

Reviewer comments:

Reviewer #1

(Remarks to the Author)

The authors have fully addressed the concerns this reviewer raised.

Reviewer #2

(Remarks to the Author)

The authors have addressed all my comments. Congratulations on a beautiful manuscript!

Reviewer #3

(Remarks to the Author)

The authors have incorporated our previous comments well, and the overall revision has improved the clarity and rigor of the manuscript. Most of the issues we raised have been adequately addressed in the revised version.

The only remaining limitation is that in vivo validation of the proposed mechanism is still relatively limited. However, considering the stated scope of the study and its focus on defining early mechanistic steps with purified components, this does not detract from the overall strength of the work.

Overall, the revision is satisfactory, and the manuscript is suitable for publication.

Re: NCOMMS-25-43530-T

Molecular mechanisms of recruitment, function and regulation of the RNA helicase UPF1 in replication-dependent histone mRNA decay

We would like to thank the reviewers for their constructive criticism and comments on our manuscript. All changes in the main text and figure legends have been highlighted in red, while additions or modifications to the supplementary information are mentioned here. We address the reviewers' comments in detail below:

Reviewer 1

Major points:

-The authors should document their cryoEM processing pipeline and the quality of their final 3D reconstruction more carefully. A detailed processing scheme outlining the individual steps should be included in the supplementary figures of the manuscript. This is especially interesting since other conformational states of the UPF1 and UPF1-SL complex might have been resolved in the data set. Additionally it is currently not clear from the included structural figures that the resolution of the cryoEM map really reaches below 3 Å as reported. The authors should therefore include a gallery of well resolved map areas with the respective model parts (below 3 Å one expects to resolve medium size amino acid side chains and RNA bases individually). If this is not possible the authors should caveat their reported resolution estimates more carefully. An map-vs-model FSC curve should be included as a supplementary figure panel and reported in table1 too. All of this is customary in reporting single particle cryoEM reconstructions.

We thank the reviewer for their further interest in our data analysis approach and have prepared an adjusted supplementary figure that describes the applied workflow accordingly (new Supplementary figure 1H). Besides overall dynamics in the SL-RNA part of the complex, no significant additional conformational states were observed (Supplementary figure 1H). We have now added representations of the density after sharpening by local resolution estimation, with clearly resolved individual nucleotides of U-SL RNA and amino acid side chains for UPF1 (new Supplementary figure 1K). Lastly, we have included a map-vs-model FSC curve in Supplementary figure 1 (new Supplementary figure 1J) and have included the map-vs-model cross-resolution in Table 1.

-The authors have carefully mapped the protein-protein interactions between UPF2 and hExo3' and between UPF1 and SLBP biochemically with several approaches. Have they attempted to predicted the structures of these minimal interacting regions and/or full lengths proteins using AlphaFold or similar machine learning-based structure prediction tools? This could potentially offer additional critical insights into the structural organisation of the histone mRNPs and could help to integrate previously published data and here reported insights into an overall structural picture further strengthening this study.

We concur with the reviewer that, in principle, harnessing the power of machine-learning-based structure prediction tools would allow us to generate a comprehensive structural picture for histone mRNA decay. We systematically carried out structure predictions on UPF2-3'hExo as well as UPF1-SLBP (with and without the SL-RNA) using Alphafold3. However, the predictions were ambiguous due to the low confidence in predicting the IDRs that mediate interactions and, therefore, could not be interpreted in a meaningful manner. This

makes it challenging to reconcile the predictions with our experimental data and integrate them into an overall model, which is why we do not include this analysis in the manuscript.

Minor points:

-FigS1 A: if this reviewer understands the experimental set up correctly the authors inhibited the activity of hExo3' by chelating divalent cations via EDTA in the reaction buffer to measure the association between SL intermediates and hExo3'. Is this correct or how did the authors inhibit RNase activity/control for robust measurements in these assays (especially with regards to the SL19 substrate)? One would expect a clarifying statement to this end in either the main text or the figure legend.

For fluorescence anisotropy experiments, 3'hExo was purified in buffer lacking magnesium chloride. Furthermore, EDTA was added to the fluorescence anisotropy buffer to chelate traces of magnesium ions that might be present in the protein. Since magnesium ions are essential for exoribonuclease activity, this ensured that the RNA substrates were not degraded by 3'hExo. The composition of the fluorescence anisotropy buffer (incorrectly stated in the previous version) has been updated and a sentence has been added to the figure legend of Supplementary figure 1 to this effect.

-FigS1 D: scale bar is missing in cryoEM micrograph; please use 100 Å to make the display comparable to the scale bar in the 2D class averages

We thank the reviewer for identifying this oversight and have added a scale bar corresponding to 50 nm in the cryoEM micrograph which is well-suited as ruler to evaluate the image.

-FigS1 G: it is highly doubtful that the quoted resolution estimates carry any meaning beyond the 1st decimal. Please correct the display and the figure legend accordingly.

Corrected, as suggested.

- table 1: please also quote the per-chain correlation coefficient between the map and the model. This is relevant since the model contains both RNA and protein moieties.

In addition to the overall CC_{mask} , we now report CC_{mask} values for the protein and RNA chains separately (CC_{mask} UPF1 and CC_{mask} U-SL RNA, respectively) in Table 1.

- The authors should explain how they filtered the cryoEM micrographs from 5,498 recorded to 5,074 used in subsequent processing. What criteria were used?

Only micrographs with an information limit estimated during CTF determination below 9 Å, maximal in-frame motion below 20 px and total motion below 100 px were considered for further analysis. We have included this information in the Methods section of the manuscript.

- It is currently unclear how many technical and/or biological replicates were used in the cross linking ms experiments reported in Figure 3. The authors should explicitly clarify this in the figure legend of Figure 3 and in the respective material and methods section.

We have modified the Methods section to state this more explicitly as follows: "The crosslinking was performed three times on the same complex reconstituted by size-exclusion chromatography (technical triplicates)." We have also added a sentence to this effect in the figure legend for Figure 3D

- Similar to the layout of the informative protein domain organisation schemes it would be helpful if the authors provided such schemes for the different SL intermediates/constructs.

We have added a schematic for the different SL variants to Figure 1A. The sequences of the different variants are detailed in Supplementary figure 1A and Figure 5E.

page 4; line 20 : "... functional mRNA decay.." it is unclear to the reader what the authors mean by "functional" here. Is there "non-functional" decay?

The term "functional" refers to the mRNA undergoing decay (as opposed to non-functional mRNA such as nonsense transcripts that are not translated). We have modified the sentence to remove this ambiguity.

page 8 line 7/8 citation/pdb code missing for SL:SLBP structure.

Modified, as suggested.

page 9 line 9/10 why do the authors perceive there to be a bigger stimulatory effect of active unwinding on hExo3' activity in cells compared to their assay set up. Isn't a greater effect through direct UPF1 recruitment via specific protein-protein interaction equally if not more plausible?

We apologise for the ambiguity of this statement. The comparison is not between the experimental setup and the scenario in cells but rather histone mRNA decay in the absence and presence of UPF1. As pointed out by the reviewer, recruitment of UPF1 has a big stimulatory effect on histone mRNA decay. We have modified this sentence to make it clearer.

page 14 line 17 there is an "in" missing; it should read "... and in previous studies"

Corrected.

page 15 line 16/17 does this sentence refer to a figure or a previous paper? please clarify and add respective reference if applicable.

This statement refers to a previous study (Meaux et al., 2018, ref. 16 in the text), for which the citation has now been added.

Material and methods "Model building..." part (starting page 24): the authors say they used both a previously published yeast as well as a human UPF1 structural model to build the UPF1 model reported here. However they do not specify for what or why the yeast model was used or how they proceeded with this. This should be clarified.

The initial placement of RNA in the UPF1-RNA binding channel was guided by the position of the poly-U RNA in the yeast Upf1-RNA structure (PDB-ID 2XZL). We have modified the methods section to include this information.

Reviewer #2

Major comments:

1) experiments in the SLBP Δ E2 cell line: this is a very interesting tool. if Antibodies for IP are available, it would be great if the authors could test by IP (WB or MS) whether the interaction between SLBP and UPF1 is lost (if possible, in the absence and presence of inhibitor of DNA replication)

Unfortunately, there are no commercial anti-SLBP antibodies that are suitable for IP and despite our best efforts, we were unable to specifically immunoprecipitate SLBP from the edited cell line.

2) Figure 5A/B: it would be great if the authors could add the N-terminal IDR (without RBD) as a control

Since the N-terminal IDR of SLBP is insoluble when expressed alone, we generated a fusion protein where the SLBP N-terminal IDR is fused to the UPF1 helicase core (UPF1-SLBP fusion) and assessed its ability to unwind the SL-RNA substrate. We observed strong inhibition of catalytic activity upon fusion of the SLBP N-IDR to UPF1, indicating that binding of this IDR to the UPF1 helicase core dampens its unwinding activity. This data has been added to Supplementary figure 5 (new Supplementary figure 5D).

Minor comments:

3) For Figure 2A / Supplementary Figure 2A: it would be helpful to see the two structures side-by-side

We have included the U-SL-RNA structure from our study in Supplementary figure 2B as a side-by-side comparison of the RNA conformations.

4) Figure 2C: please add a quantification

A quantification of the degradation assay has been added to Figure 2C.

5) Figure 3C: can the authors comment on the stoichiometry of UPF versus SLBP in their stable complex? From the gel, I am not sure it is 1:1

From extensive preparative biochemistry on UPF1-SLBP-3'hExo complexes (shown below), we conclude that that stoichiometry of the UPF1-SLBP interaction is 1:1. The sub-stoichiometric UPF1:SLBP ratio in peak 1 of Figure 3C is likely due to its overlap with peak 2 (corresponding to excess UPF1), which results in a higher amount of UPF1 in peak 1 fractions.

6) figure 5 and 6: does the fraction unwound depend on the concentration of SLBP protein? What is the molar ratio of SLBP versus UPF1 in cells?

As shown in Figure 5A, the fraction of nucleic acid substrate unwound does depend on the concentration of SLBP protein added. A 3-fold excess of SLBP over UPF1 results in a more robust inhibition of UPF1 unwinding activity. A high throughput study of protein amounts in HEK293 cells (*OpenCell, Cho et al., 2022, PMID: 35271311*) estimates a concentration of 750 nM for UPF1 and 130 nM for SLBP. However, the amount of SLBP increase by 20-fold during the S-phase of the cell cycle (*Whitfield et al., 2000, PMID: 10825184*), which would

lead to a significantly higher concentration of SLBP relative to UPF1. We make a note of this in the discussion of our manuscript.

Reviewer #3

Major comments

1. The study aims to elucidate how UPF1, SLBP, 3'hExo, and UPF2 coordinate to drive histone mRNA decay. While the structural and biochemical data convincingly show how these components interact and modulate each other's activity in vitro, the cellular context in which these interactions occur is not addressed. Given that histone mRNA decay is a rapid, tightly regulated, and cell cycle-linked process, the lack of time-resolved or phase-specific cellular experiments is a key limitation.

In particular, the authors propose that the SLBP-UPF1 interaction both recruits and regulates UPF1 activity to prevent premature histone mRNA decay. While this model is supported by in vitro structural and biochemical data, the study does not include time-resolved cellular experiments that examine how these interactions change in a physiologically dynamic context, such as during S-phase progression or in response to replication stress.

A major missing element is the role of UPF1 phosphorylation, which is a well-established regulatory mechanism controlling its helicase activity and interaction with decay cofactors. In vivo, UPF1 is phosphorylated in a cell cycle-dependent manner, which influences its recruitment, catalytic state, and decay initiation potential. However, all in vitro experiments in this study were performed with recombinant, unphosphorylated UPF1, thereby excluding this essential regulatory layer.

As the reviewer points out, the cellular context leading to histone mRNA decay at the end of the S-phase of the cell cycle is very complex. It involves integration of signals that mark the end of the S-phase with signals derived from translation termination on the histone mRNA. We have previously carried out extensive time-resolved and phase-specific cellular experiments to develop a model for how histone mRNA is degraded (*reviewed in Marzluff and Koreski, 2017, PMID: 28867047*). In this manuscript we focus on how the critical components involved in the initial steps of histone mRNA decay (UPF1, SLBP and 3'hExo) interact with each other and act in conjunction to mediate degradation.

Our previous studies in mammalian cells have demonstrated that UPF1 is recruited to the 3'-UTR of histone mRNAs along with its activator, Smg1, *in vivo* within minutes after inhibition of DNA replication (*Kaygun et al., 2005, PMID: 16086026, Meaux et al., 2018, PMID: 30397106*). In lieu of activation by SMG1 in cells, we used an "active" form of UPF1 (UPF1-Hel) for our studies. Although the interaction between UPF1 and SLBP was previously described (*Choe et al., PMID: 25016523*), the present study demonstrates for the first time that UPF1 directly interacts with SLBP bound to the 3' end of histone mRNA. We identified the region on SLBP required for binding to UPF1 as well as the region of UPF1 that interacts with SLBP. The SLBP proteins were expressed and purified using a baculovirus expression system and are phosphorylated (Supplementary figure 3B). We showed that the UPF1-SLBP interaction is independent of the phosphorylation state of SLBP as both phosphorylated and unphosphorylated SLBP are co-precipitated with UPF1 (Figures 3A and 3B). To address if phosphorylation of UPF1 has an impact on its interaction with SLBP, we expressed and purified His-Flag-UPF1fl from HEK293 cells and tested its binding to SLBPfl. As a control, we treated the cell lysate with calf intestinal phosphatase (CIP) and analysed the UPF1-SLBP interaction. UPF1fl expressed in HEK293 cells showed robust binding to SLBP, both with and without CIP treatment, indicating that phosphorylation of UPF1 does not affect its binding to SLBP. This is consistent with the observation that binding to SLBP is mediated by the UPF1

helicase core that is not phosphorylated. This data has been added to Supplementary figure 3 (new Supplementary figure 3C).

2. *Figures 1B and 5A–C present unwinding kinetics that are central to the paper’s mechanistic model — namely, that UPF1-mediated stem-loop unwinding is modulated by SLBP and its domains. However, the interpretation of these data lacks statistical rigor, and the trends reported are not sufficiently substantiated with quantitative analysis.*

In Figure 1B, the unwinding efficiency of SL RNA appears to be greater than that of the linear RNA. Yet it is unclear whether this difference is statistically significant or simply due to experimental variation.

In Figures 5A–C, SLBP inhibits UPF1 unwinding via its N-terminal IDR, while SLBP-RBD shows no effect on linear RNA (5B) but unexpectedly slows early unwinding and enhances it later on SL RNA (5C), a biphasic effect that requires clearer explanation and statistical validation.

We have carried out unpaired t-tests between the end points of the unwinding reactions (60 min) in Figures 1B, 5A and 5B and report statistical significance (derived from p-values). This analysis shows that there is no significant difference between the linear and SL RNA with UPF1 alone (Figure 1B). For the data presented in Figures 5C, 6A and 6B which highlight differences in the early stages of the unwinding reaction, we compare the fraction of substrate unwound after 10 min and report statistical significance. The corresponding figures and figure legends have been modified to highlight the statistical significance. The p-values are reported in the respective supplementary figures.

The unwinding reaction described in this study appear to be a two-phase reaction, with a rapid unwinding phase (prominent in the early time points) and a slower unwinding phase (prominent in the later time points). As pointed out by the reviewer, addition of SLBP-RBD to the SL RNA unwinding reaction slows down early unwinding but does not impact the later phase. The experimental data were fit to a two-phase decay model and first-order rate constants, k_{fast} and k_{slow} , were derived for both phases (shown in the respective supplementary figures). We propose that the rate constants for unwinding of the SL RNA substrate in presence of SLBP-RBD and SLBPfl reflect their impact on the early and late stages of unwinding, respectively.

Minor comments

1. *Figure 1D/E: The rationale for introducing a polyuridine (polyU) sequence upstream of the SL in the RNA construct should be clarified.*

A poly-U sequence was added to the 5'-overhang of the RNA substrate used for cryoEM studies to prevent the formation of secondary structure that could impact the binding of UPF1 to this stretch of RNA. We have clarified this in the manuscript.

2. *Figure 3A: The role of GST-GYF as a negative control in the pulldown assay should be explained.*

GST-GYF was used as a negative control to mimic hydrophobic interaction interfaces that could non-specifically capture aggregated SLBP proteins. We have added this explanation to the methods section of the manuscript.

3. *Figure 3C: The interpretation of peak 4 as aggregated SLBP-N needs clarification. Since both SLBP-N and UPF1 show smearing, it is recommended to distinguish specific aggregation from nonspecific background.*

Although SLBP that is not bound to SL-RNA elutes within the included volume of the size-exclusion column, it does not bind UPF1, suggesting that it is not in the right conformation

to mediate protein-protein interactions. The results presented here as well as other experiments not included in this study show that SLBP has a strong tendency to aggregate in the absence of SL RNA. This is distinct from a slight precipitation observed in many complex eukaryotic proteins upon freeze-thawing, which can be removed by high-speed centrifugation. We mention this property of SLBP in the manuscript.

4. Figure 3D: Clearly labeling the domain architecture of both UPF1 and SLBP in the figure would improve interpretability of the crosslinking results.

We have modified Figure 3D to include labels for domains 1B and 1C of UPF1 (in addition to RecA1 and RecA2 domains) and the RBD of SLBP.

5. Figure 4C: The authors conclude that deletion of SLBP residues 29–51 impairs UPF1 recruitment. However, this region might also mediate interactions with other unknown cellular cofactors. The possibility of indirect effects should be acknowledged.

We concur with the reviewer and have therefore included the following sentence in the results section where Figure 4C is described: “It is important to mention that the N-terminal IDR of SLBP also engages other factors, such as those involved in nuclear export and translation of histone mRNA, although these interactions occur at distinct points of a histone mRNA transcript’s lifetime in the cell. It is possible that interactions of SLBP N-IDR with yet unknown cellular factors might further influence the rate of histone mRNA decay.”